# Untraceable DeepFakes via Traceable Fingerprint Elimination

**Jiewei Lai**[1], **Lan Zhang**[1,2*] , **Chen Tang**[1], **Pengcheng Sun**[1],**Xinming Wang**[1], **Yunhao Wang**[3]

[1]University of Science and Technology of China, [2]Institute of Artificial Intelligence,
Hefei Comprehensive National Science Center , [3]Lenovo Research

`jw_lai@mail.ustc.edu.cn,zhanglan@ustc.edu.cn,{chentang1999,`
`speical0806,xinmingwang}@mail.ustc.edu.cn,wangyh43@lenovo.com`

## ABSTRACT

Recent advancements in DeepFakes attribution technologies have significantly enhanced forensic capabilities, enabling the extraction of traces left by generative models (GMs) in images, making DeepFakes traceable back to their source GMs. Meanwhile, several attacks have attempted to evade attribution models (AMs) for exploring their limitations, calling for more robust AMs. However, existing attacks fail to eliminate GMs' traces, thus can be mitigated by defensive measures. In this paper, we identify that untraceable DeepFakes can be achieved through a multiplicative attack, which can fundamentally eliminate GMs' traces. Therefore, by leveraging the structural prior from content-coupled fingerprints, we design a multiplicative attack framework that instills an explicit inductive bias into the adversarial model, guiding it to eliminate fingerprints within DeepFakes, thereby evading AMs even enhanced with defensive measures. This framework trains the adversarial model solely using real data, applicable for various GMs and agnostic to AMs. Experimental results demonstrate the outstanding attack capability and universal applicability of our method, achieving an average attack success rate (ASR) of 97.08% against 6 advanced AMs across 12 GMs. Even in the presence of defensive mechanisms, our method maintains an ASR exceeding 72.39%. Our work underscores the potential challenges posed by multiplicative attacks and highlights the need for more robust AMs.

## 1 INTRODUCTION

With the rapid development of GMs, such as generative adversarial networks (GANs) (Karras et al., 2018; Miyato et al., 2018; Binkowski et al., 2018) and diffusion models (DMs) (Rombach et al., 2022), creating realistic and diverse high-quality images is becoming easily accessible. However, this progress also enables misuse, leading to issues such as misinformation and intellectual property infringement. Therefore, dedicated research efforts are being paid to forensics, such as DeepFakes detection for authenticity identification (Wang et al., 2025).

DeepFakes attribution, a method that surpasses DeepFakes detection by identifying both the authenticity of an image and the specific model or type of models used to generate it, is a promising approach for enhancing accountability among malicious content creators. This technology captures distinctive traces left by GMs within images, known as the model fingerprint, thereby attributing generative content to its source model. Additionally, it supports intellectual property protection by identifying unauthorized use of copyrighted models or their generative content.

With significant advances in DeepFakes attribution, research into the vulnerability of AMs, known as attribution attacks, has emerged to explore their limitations, thereby fostering more robust attribution methods and effective countermeasures. Current attacks evade AMs by adding perturbations into images, called additive attacks, demonstrating considerable attack performance. However, our analysis and preliminary experiments reveal a fundamental flaw: they are easy to defend against because they fail to eliminate the fingerprints that are essential for attribution. This implies that while

---

*Corresponding author.

existing methods can temporarily circumvent attribution, their inability to eliminate fingerprints renders them inherently fragile and circumventable due to persistent forensic traces.

Therefore, this paper further explores attack methods against attribution methods, developing an attack strategy to achieve untraceable DeepFakes via fingerprint elimination that not only evades AMs but also circumvents defense mechanisms. To achieve this objective, three critical issues must be addressed: 1) The trade-off between fingerprint elimination and visual imperceptibility. While more extensive modifications generally enhance attack performance, they can also compromise image quality. Moreover, failing to eliminate the fingerprint undermines the effectiveness of the attack. 2) The broad spectrum of GMs and their diverse fingerprint characteristics present significant complexity. It is impractical to design a model-specific method, therefore, ensuring the universality of our approach enhances its applicability to DeepFakes generated by various GMs without requiring customization. 3) Additionally, the adversary often lacks knowledge of the attribution mechanisms in practical scenarios. Thus, designing a model-agonist attack that is independent of specific AMs is essential for ensuring its effectiveness in evading various unknown AMs.

To address these challenges, we propose a universal and black-box attack strategy targeting AMs and defensive mechanisms by fingerprint elimination. By leveraging the structural prior that model fingerprints are content-coupled modulations, we identify the multiplicative attack that can eliminate traceable fingerprints while preserving perceptual integrity through an adversarial matrix. We rigorously prove the existence of multiplicative adversarial matrices and demonstrate that defending against this attack is fundamentally constrained by statistical limits: 1) Without paired clean and adversarial images, inverting the multiplicative attack is non-identifiable; 2) Even with paired data, achieving low estimation error requires a prohibitively large sample size, rendering practical defense unreliable. Subsequently, we design a universal and black-box multiplicative attack framework that leverages this structural prior to instill an inductive bias into the adversarial model, guiding it to learn to effectively eliminate fingerprints within DeepFakes while preserving the visual fidelity. This framework constructs the model using exclusively real data without requiring any DeepFakes or GMs. Specifically, this framework comprises three modules: 1) Data synthesis: employs sampling and transformation units to create synthetic data from real data, mimicking the characteristics of DeepFakes without access to GMs. 2) Model Construction: trains the model to eliminate artificial fingerprints within synthetic images through explicit joint optimization of both visual fidelity and fingerprint elimination, thereby enabling genuine fingerprint elimination rather than mere obscuring. 3) Fingerprint Elimination: given DeepFakes generated by any GMs, the resulting model effectively serves as the adversarial matrix to eliminate fingerprints rather than merely obscuring, thus evading unknown AMs even with the presence of the defensive mechanisms.

Our main contributions are summarized as follows:

- We reveal that current attacks against AMs are constrained to additive perturbations through analysis and preliminary experiments. This additive nature inherently preserves model-specific fingerprints, rendering them highly susceptible to effective defensive mechanisms.
- We theoretically identify that the multiplicative attack can provably eliminate GMs' fingerprints by leveraging their content-coupled modulations, and prove this attack is statistically non-invertible, rendering it intrinsically evasive against AMs, even under defenses.
- We propose a universal and black-box multiplicative attack method which instills the inductive bias into the adversarial model, enabling it to effectively eliminate traceable fingerprints while preserving visual fidelity without requiring any DeepFakes or access to AMs.
- We experimentally validate our method's effectiveness against 6 advanced AMs across 12 GMs with an ASR of 97.08%, surpassing SOTA methods. Crucially, it achieves an ASR of more than 72.39% against defensive measures. Quantitative analysis further confirms that our method can effectively eliminate fingerprints and validate its multiplicative nature.

## 2 RELATED WORKS

### 2.1 DEEPFAKES ATTRIBUTION

DeepFakes attribution technologies focus on extracting a unique fingerprint left by GMs within DeepFakes to determine its source model. The pioneering work introduced model fingerprints and

designed the AttNet framework to trace the source model. Building upon this foundation, subsequent research studies actively inserted transferable fingerprints into GMs, thereby enabling the decoupling of the fingerprint from the generated content (Yu et al., 2021; 2022). To enhance the capabilities of AMs in identifying unseen GMs, researchers explored DeepFakes attribution under an open-set setting (Girish et al., 2021; Yang et al., 2023). Instead of attributing images to specific models, these studies aimed at architecture-level attribution, attributing images back to their source architectures (Frank et al., 2020; Yang et al., 2022; Bui et al., 2022; Asnani et al., 2023). DCT revealed that architecture fingerprints cause severe artifacts in the frequency domain and performed attribution within this domain (Frank et al., 2020). DNA-Det identified the global consistency of architectural fingerprints and developed a patch-wise contrastive learning-based framework for attribution (Yang et al., 2022). Meanwhile, some studies achieved this goal through a mixing representation strategy and reverse engineering, respectively (Bui et al., 2022; Asnani et al., 2023). Recently, the rise of DMs spurred research into attributing images generated by DMs. For example, the image and its description are simultaneously utilized for DeepFakes detection and attribution (Sha et al., 2023). Besides, reconstruction errors were used to infer the source model, as well-reconstructed images are likely generated from the inspected model (Wang et al., 2024; Laszkiewicz et al., 2024).

## 2.2 Anti Forensics

DeepFakes forensics are critical for curbing misuse and establishing responsibility, driving many efforts devoted to exploring the vulnerabilities of existing forensic approaches, thus promoting more advanced technologies, including DeepFakes detection and attribution. Early anti-forensic work utilized adversarial examples like FGSM (Goodfellow et al., 2015), PGD (Madry et al., 2018) to add perturbations into DeepFakes for evading detectors (Neekhara et al., 2021; Liao et al., 2021). DiffAttack subsequently used DMs to generate highly transferable perturbations (Chen et al., 2024a). Similarly, imperceptible semantic level perturbations were designed through latent space optimization (Meng et al., 2024). Some studies (Wesselkamp et al., 2022; Liu et al., 2023) evaded detectors by reducing detectable artifacts rather than adding perturbations. For instance, FakePolisher learned real representation from real images to construct a dictionary for reducing artifacts (Huang et al., 2020). StealthDiffusion (Zhou et al., 2024) achieved this by optimizing both the latent space and the frequency domain, enabling images indistinguishable from real images. However, research on the vulnerability of AMs remains limited. Specifically, the vulnerability faced with transformation-based methods like compression were explored (Yu et al., 2019; Yang et al., 2022). Transferable adversarial samples were also attempted to attack AMs (Wu et al., 2024). TraceEvader, a universal attack method by inserting perturbations into high frequency information and blurring low frequency information of images, thus confusing traceable fingerprints to evade AMs (Wu et al., 2024).

Although existing methods achieve high ASR, they fail to eliminate the underlying model fingerprints within DeepFakes. In this paper, we propose a universal, black-box attack against AMs that fundamentally eliminates these fingerprints. Consequently, our method is significantly more challenging to defend against, as demonstrated by both theoretical analysis and extensive experiments.

## 3 Preliminary Statement

### 3.1 Threat Model

**Defender's Goals and Capabilities:** The goal of the defender is to trace the source model of Deep-Fakes, determining which model instance or type of architecture the image is from. The defender 1) trains attribution models with available clean DeepFakes and 2) attempts to alleviate the vulnerability of AMs through defensive strategies such as adversarial training. The defender may also collect or create some adversarial DeepFakes for enhancing attribution robustness.

**Adversary's Goals and Capabilities:** The goal of the attack is to generate DeepFakes and enhance untraceability to evade AMs even under defensive measures, thereby avoiding responsibility. The attacker 1) has full ownership of GMs and knowledge of training data, 2) has no access to or information about any AMs, and 3) aims to develop a universal and black-box attack that eliminates model fingerprints, rendering untraceable DeepFakes without altering content perceptibly.

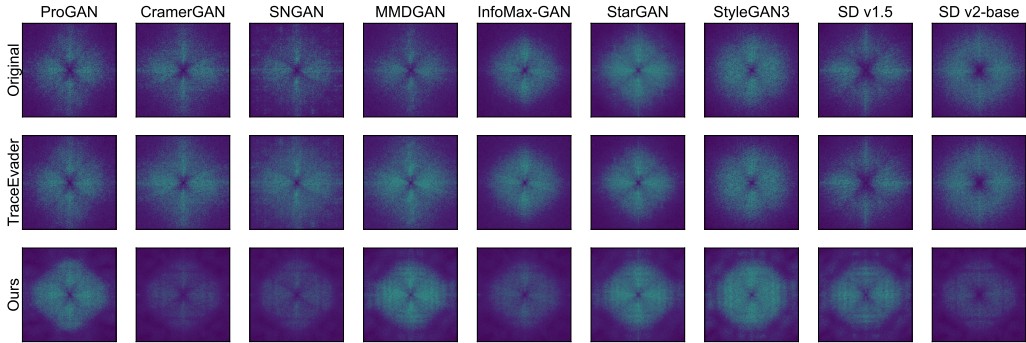

Figure 1: Spectral analysis of (top to bottom): original, additive attack images, and images processed with our method. See A.1.2 (Figures 6,7,8) and A.4.2 (Figure 15) for more.

## 3.2 PROBLEM FORMULATION

Inspired by camera fingerprint studies (Qian et al., 2023), an image $x$ generated by a model $\mathcal{M}$ can be modeled as: $x = x^0 + x^0 f_{\mathcal{M}} + \Theta$, where $x^0$ represents the visual content of the image, $f_{\mathcal{M}}$ denotes the model-specific fingerprint left by model $\mathcal{M}$, and $\Theta$ captures other noise components. For brevity, we will use $f_{\mathcal{M}}$ to refer to the entire fingerprint term. Based on this formulation, we provide definitions for DeepFake attribution and attribution attack as follows:

**Definition 1** (DeepFake Attribution). *Let $\mathcal{F} : \mathbb{R}^{\mathcal{CHW}} \to \mathcal{P}(\mathcal{M})$, where $\mathcal{P}(\mathcal{M})$ denotes the power set of all possible GMs, be an attribution model that identifies the source GMs $\mathcal{M}$ responsible for generating $x$. $\mathcal{F}$ aims to extract $f_{\mathcal{M}}$ from $x$ through*

$$\mathcal{F}(x) \mapsto f_{\mathcal{M}} \quad s.t. \quad \|\mathcal{F}(x) - f_{\mathcal{M}}\| < \epsilon,$$

*where $\epsilon$ is a tolerance threshold for estimation errors.*

**Definition 2** (Attribution Attack). *Let $\mathcal{T} : \mathbb{R}^{\mathcal{CHW}} \to \mathbb{R}^{\mathcal{CHW}}$ be an attack model that generates an adversarial image $\mathcal{T}(x)$ to mislead $\mathcal{F}$ while preserving visual fidelity, that is*

$$\|\mathcal{F}(\mathcal{T}(x)) - \mathcal{F}(x)\| \geq \epsilon \text{ and } \mathcal{D}(\mathcal{T}(x), x) \leq \Delta,$$

*where $\mathcal{D}(\cdot, \cdot)$ is a metric bounding the visual distortion and $\Delta$ bounds the allowable perturbation.*

This paper aims to design an attack method that is applicable to images generated by any GMs (universal) and capable of misleading all AMs without prior knowledge (black-box).

## 3.3 EXISTING ATTACK ANALYSIS

In this section, we establish that existing attacks are fundamentally additive in nature and, crucially, fail to eliminate the underlying model fingerprint, rendering them inherently vulnerable to defense.

**Definition 3** (Additive Attack). *Let $\mathcal{T}_{add} : \mathbb{R}^{\mathcal{CHW}} \to \mathbb{R}^{\mathcal{CHW}}$ be an additive attack model that generates adversarial images through inserting the perturbation into images: $\mathcal{T}_{add}(x) = x + p$.*

In this context, a clean image $x$ generated is attacked as $\mathcal{T}_{add}(x) = x^0 + f_{\mathcal{M}} + p + \Theta$. The success of additive attack methods lies in their ability to add a carefully crafted perturbation $p$ that obscures the fingerprint within DeepFakes, thereby misleading AMs. It is important to note that this merely confuses the fingerprint, increasing the difficulty of extracting the fingerprint, however, the fingerprint itself remains intact within DeepFakes.

We identify that existing attack methods are additive (Detailed analysis details in the Appendix A.1.1), therefore, they are easily defended against as they fail to eliminate the fingerprint within DeepFakes. Our defensive experiments and frequency analysis (Details in the Appendix A.1.2) both reveal the limitations of them: *1) easily defended against and 2) fail to eliminate the fingerprint*:

*1) Defensive Evaluations:* We employ adversarial training to enhance the DNA-Det (Yang et al., 2022) and evaluate the effectiveness of additive attacks against defensive mechanisms. Experimental results reveal that additive attacks can be easily and effectively countered by a defensive strategy. As summarized in Fig 3, the performance of all tested attacks experiences a significant decline

following adversarial training. For instance, TraceEvader's ASR drops from 98.28% to 25.10% against the defensive mechanism. By incorporating adversarial training, defenders can significantly enhance the robustness of AMs to handle additive attacks.

*2) Frequency Analysis:* Model fingerprints exhibit pronounced characteristics in the frequency domain, manifesting as distinct patterns (Frank et al., 2020). Our frequency analysis provides evidence of the persistence of these fingerprints within adversarial images. As illustrated in Figure 1, images attacked by TraceEvader (Wu et al., 2024) still exhibit high similarity in the frequency domain with their original counterparts, demonstrating that additive attacks fail to eliminate the fingerprint.

Therefore, merely confusing the fingerprint is insufficient to truly evade AMs. Conversely, we argue that true non-traceability can only be achieved by eliminating the fingerprint within DeepFakes.

## 4 METHODOLOGY

In this section, we first prove the existence of multiplicative attacks that provably eliminate GMs' fingerprints and show that it is statistically non-invertible, rendering defense fundamentally limited. We then present a universal black-box framework that uses only real data to synthesize fingerprint-mimicking data and train an adversarial model to eliminate fingerprints across diverse GMs.

### 4.1 MULTIPLICATIVE ATTACK

The principle of attribution methods is the extraction of the fingerprint $f_{\mathcal{M}}$ from DeepFakes. Therefore, we argue that true untraceability requires eliminating these fingerprints at their source rather than merely obscuring them. To achieve this, we propose a multiplicative attack:

**Definition 4** (Multiplicative Attack). *Let $\mathcal{T}_{mul} : \mathbb{R}^{CHW} \to \mathbb{R}^{CHW}$ be a multiplicative attack model that generates adversarial images with an adversarial matrix $W$: $\mathcal{T}_{mul}(x) = x \odot W$.*

*Structural Prior from Content-Coupled Fingerprints :* This formulation leverages a critical property of generative fingerprints: GMs' fingerprints are not independent noise but are intrinsically coupled with the image content. They arise from content-dependent operations such as up-sampling, which manifest as structured modulations (e.g., grid-like patterns) (Odena et al., 2016; Frank et al., 2020).

*The Inductive Bias for Elimination:* The multiplicative attack leverages this structural prior as an explicit inductive bias, directly disrupting this modulation mechanism through a multiplicative operation with an adversarial matrix. Specifically, in this context, a clean image $x$ is attacked as: $\mathcal{T}_{mul}(x) = x^0 \odot W + f_{\mathcal{M}} \odot W + \Theta$. Here, $f'_{\mathcal{M}} = f_{\mathcal{M}} \odot W$ is the altered fingerprint, by optimizing the adversarial matrix $W$, making $f'_{\mathcal{M}}$ distinct from the original $f_{\mathcal{M}}$. the source GMs can not be traced because $f'_{\mathcal{M}} \neq f_{\mathcal{M}}$. By eliminating the original fingerprint, this attack removes all residual source-specific information, rendering attribution impossible even under advanced defenses.

*Theoretical Existence of Adversarial Matrix:* We prove that there always exists a multiplicative adversarial matrix $W$ capable of constructing an adversarial image as $x' = x \odot W$ to evade $\mathcal{F}$ while preserving visual fidelity. This is formally established in Theorem 1 and proved in Appendix A.2.

### 4.2 DEFENSE DIFFICULTY ANALYSIS

We show that the multiplicative attack is statistically non-invertible, making inversion and approximate inversion defenses infeasible, rendering it intrinsically evasive against both AMs and defenses:

*1) Inversion Defense:* Let $x' = P(x \odot W) + \eta$, where $P$ denotes standard pre-processing and $\eta$ represents noise. Without paired supervision, inverting $x$ from $x'$ is non-identifiable. In practice, access to such paired samples is unrealistic. Even with $N$ paired samples and a per-pixel Gaussian model $x'_j = x_j W_j + \eta_j$, $\eta_j \sim \mathcal{N}(0, \sigma^2)$, any unbiased estimator satisfies $\mathrm{Var}(\widehat{W_j}) \geq \sigma^2 / (N \, \mathbb{E}[x_j^2])$. Consequently, achieving MSE $\leq \varepsilon^2$ requires $N \gtrsim \sigma^2 / (\varepsilon^2 \, \mathbb{E}[x_j^2])$. Details in Appendix A.2.2.

*2) Approximate Inversion :* Using a neural network to approximate $W^{-1}$ also requires a large number of image pairs, which is impractical. Moreover, using a network to invert images imprints its fingerprints onto the recovered content, further degrading defense efficacy. Our experiments show that even with simultaneous access to both clean and adversarial images, defenders still cannot reliably attribute the source of DeepFakes. Details in Section 5.3 and Appendix A.4.2.

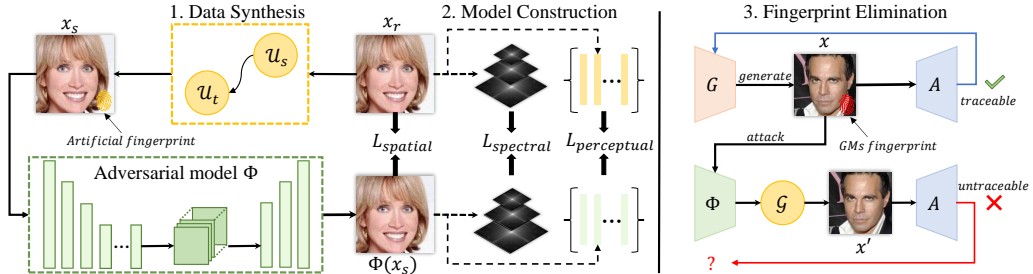

Figure 2: Overview of our multiplicative attack framework. (1): Data synthesis via sampling $\mathcal{U}_s$ and transformation $\mathcal{U}_t$ units to generate fingerprint-mimicking data $x_s$. (2): Adversarial model $\Phi$ trained with a multi-domain loss. (3): $\Phi$ attacks a DeepFakes $x$ into $x'$, which evades AMs $A$ while preserving realism. $G$ denotes the source GMs; $\mathcal{G}$ is a post-processing smoothing operator.

## 4.3 MULTIPLICATIVE ADVERSARIAL ATTACK FRAMEWORK

### 4.3.1 PARAMETERIZING THE MULTIPLICATIVE MATRIX

While Definition 4 and Theorem 1 establish the existence of a multiplicative matrix $W$ that eliminates fingerprints, directly optimizing such a matrix faces two fundamental challenges: *1) Computationally Infeasible*: It is impractical to optimize and store the multiplicative matrix for every potential input. AMs are often inaccessible for optimization, and huge storage requirements render deployment physically impossible. *2) Limited Generalization*: a fixed $W$ matrix optimized on one image would lack generalization capability as fingerprint patterns vary across diverse GMs.

To overcome these limitations, we propose parameterizing $W$ as an input-dependent function $W(x)$ and modeled via a neural network $\Phi$ that implements the multiplicative attack: $\Phi(x) = x \odot W(x)$. By modeling $W(x)$ with a compact neural network, we avoid the infeasible storage (only requiring fixed parameters) and computational cost per image implied by a fixed-matrix approach. The input-dependent nature of $\Phi(x)$ inherently enables generalization across diverse inputs, resolving the fixed-matrix limitation. Critically, learning $\Phi(x) = x \odot W(x)$ preserves the multiplicative attack structure while ensuring numerical stability, as validated quantitatively in Section 5.4.

### 4.3.2 FRAMEWORK DESIGN

**Objective.** The primary objective of this framework is to construct an adversarial model that serves as the adversarial matrix to eliminate fingerprints within DeepFakes, thus evading AMs, even under enhanced defenses. This model must satisfy: *1) Imperceptibility-Effectiveness Trade-off:* Maintaining visual fidelity while ensuring effective fingerprint elimination in both pixel and frequency domains.*2) Universality:* Eliminating fingerprints across diverse GMs without model-specific customization. *3) Black-Box:* Effectively against various AMs without prior knowledge. Therefore, our framework is explicitly designed to embed a targeted inductive bias into the adversarial model, guiding it to learn how to effectively eliminate fingerprints by leveraging their content-coupled structural prior. This design enables end-to-end construction without relying on any DeepFakes, GMs, or access to specific AMs, ensuring universal applicability across diverse GMs.

**Overview.** As illustrated in Figure 2, our framework implements an end-to-end pipeline with tightly coupled modules that collectively achieve fingerprint elimination. Our framework proceeds in three sequential steps: First, the data synthesis module generates fingerprint-mimicking images by applying sampling and transformation operations to real data, effectively simulating the artifacts left by diverse GMs without requiring access to any GM. Second, the model construction module trains an adversarial network on these synthetic pairs to eliminate artificial traces through joint optimization in perceptual, spatial, and frequency domains. Crucially, each loss term explicitly targets a fingerprint-carrying characteristic: perceptual loss preserves semantics while enabling modification, spatial loss and spectral loss suppresses low-level pixel artifacts and high-frequency traces respectively, ensuring genuine fingerprint elimination rather than mere visual fidelity. Finally, the trained model functions as a parameterized multiplicative operator that intrinsically removes traceable fingerprints from any DeepFake, producing untraceable outputs while maintaining perceptual quality.

**Insight.** The following motivation drives us to mimic the fingerprints of GMs: *1) Sampling Operations in GMs*: GMs primarily rely on down/up-sampling operations to generate images, such as nearest neighbor sampling, which introduces grid-like patterns in both the spatial and frequency domains (Odena et al., 2016; Frank et al., 2020). These patterns are inherent artifacts of the sampling processes used by GMs. *2) Image Transformation Similarities*: Certain transformation operations exhibit characteristics similar to those found in GMs. For example, blurring and adding noise using kernels closely resemble the convolution computations (Yang et al., 2022). These transformations share spatial properties with the operations within GMs. Therefore, the traces left by sampling and transformation operations exhibit properties that are analogous to the fingerprints of GMs.

**Data Synthesis.** To effectively mimic the fingerprints of GMs, we design sampling and transformation units within the data synthesis module, as illustrated in Figure 2. This module first applies the sampling unit $\mathcal{U}_s$ to real images, followed by the transformation unit $\mathcal{U}_t$ to yield synthetic images:

*1) The Sampling Unit $\mathcal{U}_s(\cdot)$* employs three sampling techniques: nearest-neighbor, bilinear, and bicubic interpolation. Given a real image $x_r \in \mathbb{R}^{\mathcal{CHW}}$, it is first down-sampled to $x_{down} \in \mathbb{R}^{\mathcal{C'H'W'}}$, where $\mathcal{C}' = \mathcal{C}, \mathcal{H}' = \mathcal{H}/2$, and $\mathcal{W}' = \mathcal{W}/2$. And then the image is up-sampled back to its original dimensions, resulting in $x_{up} \in \mathbb{R}^{\mathcal{CHW}}$. Specifically, the sampling unit stochastically applies down/up-sampling with probability $p_1$: $x_{up} = \mathcal{U}_s(x_r, s_{down}, s_{up}, p_1)$, where $s_{down}, s_{up}$ are selected randomly to introduce diverse spatial artifacts and enhance robustness.

*2) The Transformation Unit $\mathcal{U}_t(\cdot)$* incorporates a series of image transformation techniques to introduce diverse and realistic variations: (a) Gaussian noise sampled from $\mathcal{N}(0, \sigma^2)$, where $\sigma^2$ is randomly selected from [5.0, 20.0]; (b) Gaussian filtering with a kernel size randomly chosen from $\{1,3,5\}$; (c) Randomly crops with an offset between 5-20% of the image lengths; (d) JPEG compression using a quality factor randomly sampled from [10,75]; (e) Relighting (adjusts brightness, contrast, and saturation) with random factors from [0.5, 1.5]; (f) Combination processes in the following order: relight, cropping, blurring, compression and adding noise. one of the above operation $t$ is randomly selected and applied with probability $p_2$: $x_s = \mathcal{U}_t(x_{up}, t, p_2)$.

By utilizing sampling and transformation operations, we can effectively simulate the properties of GM-generated data without access to specific GMs, avoiding limitations to any particular GM.

**Model Construction.** The adversarial model $\Phi$ is trained end-to-end on real/synthetic image pairs $(x_r, x_s)$ to eliminate artificial fingerprints within synthetic images while preserving perceptual fidelity. $\Phi$ comprises an encoder-decoder architecture. The encoder includes 3 convolutional layers and 5 residual layers to extract feature maps. The decoder utilizes 2 up-sampling layers and a convolutional layer to reconstruct images. More details in the Appendix A.3 and Table 2. The model is optimized to target both visual fidelity and fingerprint elimination explicitly:

*1) Fidelity Preserving*: To ensure visual and semantic similarity between $\Phi(x_s)$ and $x_r$, we employ a perceptual loss function with a pretrained 16-layer VGG network as the backbone to extract high-level features. The perceptual loss is defined as: $\mathcal{L}_{perceptual} = \sum_{i \in F} w_i \|f^i_{\Phi(x_s)} - f^i_{x_r}\|_2$, where $f^i_x$ denotes the feature extracted from the selected layer $F = \{f^{i_1}, f^{i_2} \dots f^{i_j}\}$ of the perceptual network for image $x$, and $w_i = \frac{1}{|F|}$ represents the weight.

*2) Fingerprint Elimination*: Given that synthesis operations and GMs introduce content-coupled artifacts in both spatial and frequency domains, we design a spatial loss to remove low-level artifacts within the pixel domain and a multi-scale spectral loss for frequency domain optimization. Specifically, the spatial loss is formulated as $\mathcal{L}_{spatial} = \|\Phi(x_s) - x_r\|_2$. For frequency domain artifacts elimination, we apply the Fourier Transform to obtain the frequency representation of a scaled image $x_s$ with scale factor $s \in S = \{1, 0.5, 0.25\}$, calculating the magnitude of the frequency components, and applying logarithmic scaling to stabilize numerical computations. And a small constant $\varepsilon$ is added to prevent taking the logarithm of zero values. The final spectral loss is formulated as $\mathcal{L}_{spectral} = \sum_{s_i \in S} w_i \|\mathcal{L}(\Phi(x_s), s_i) - \mathcal{L}(x_r, s_i)\|_1$, where $\mathcal{L}(x, s_i) = log(|fft(x^{s_i})| + \varepsilon), w_i \in \{0.5, 0.3, 0.2\}$ represents the weight. The total training loss function is formulated as: $\mathcal{L}_{total} = \beta_1 \mathcal{L}_{perceptual} + \beta_2 \mathcal{L}_{spatial} + \beta_3 \mathcal{L}_{spectral}$.

By training the model with a multi-domain optimization strategy, we instill in the attack model $\Phi$ an inductive bias for fingerprint elimination, enabling it to remove artificial fingerprints while preserving visual fidelity, thereby realizing the core principle of multiplicative attack: eliminating the fingerprint left by GMs rather than merely obscuring.

Table 1: Comparison of ASR (%) across various AMs. The best and second-best results are marked in **bold** and underlined. The symbol '-' within the table indicates not applicable or limited by computational overhead. The performance of AMs on clean samples is presented in Table 4.

| | DNA-Det | AttNet | DCT | Reverse | POSE | LTracer | Average | SSIM | LPIPS |
|---|---|---|---|---|---|---|---|---|---|
| PGD | - | 99.98 | 66.16 | 59.77 | 6.75 | - | 58.17 | 0.912 | 0.401 |
| BIM | - | 0.75 | 47.25 | 43.78 | 3.40 | - | 23.79 | 0.910 | 0.401 |
| MIFGSM | - | 8.90 | 80.25 | 67.58 | 1.30 | - | 39.51 | 0.911 | 0.401 |
| DiffAttack | - | **100.00** | 66.75 | 25.40 | 56.8 | - | 62.24 | 0.962 | 0.095 |
| Transformation | 43.70 | 99.96 | 61.21 | 54.89 | 47.01 | 98.79 | 67.60 | 0.941 | 0.151 |
| FakePolisher | 75.00 | 40.95 | 53.70 | **90.33** | **95.85** | - | 71.17 | 0.994 | 0.067 |
| Regeneration | 93.30 | 99.95 | 84.46 | 81.05 | 39.71 | 0.00 | 78.60 | 0.912 | 0.210 |
| TraceEvader | 98.28 | 65.45 | 92.40 | 88.80 | 86.81 | 90.89 | 87.11 | **0.995** | **0.038** |
| Ours | **98.56** | **100.00** | **100.00** | 89.54 | 95.52 | **98.89** | **97.08** | 0.963 | 0.093 |
| Ours (w/o $\mathcal{U}_s$) | 98.10 | 100.00 | 100.00 | 84.75 | 93.76 | - | 95.32 | 0.970 | 0.076 |
| Ours (w/o $\mathcal{U}_t$) | 91.69 | 100.00 | 100.00 | 81.48 | 95.83 | - | 93.80 | 0.968 | 0.083 |
| Ours (w/o GBMS) | 86.15 | 100.00 | 100.00 | 67.83 | 95.14 | - | 89.82 | 0.965 | 0.102 |
| Ours (w/o $\mathcal{L}_{spatial}$) | 86.34 | 100.00 | 100.00 | 87.80 | 96.89 | - | 94.21 | 0.964 | 0.102 |
| Ours (w/o $\mathcal{L}_{spectral}$) | 92.45 | 100.00 | 61.16 | 50.04 | 97.92 | - | 80.31 | 0.950 | 0.089 |

**Fingerprint Elimination.** After training, given a clean and traceable generative image $x$, the attack model $\Phi$ eliminates the fingerprints left by the corresponding source GMs, thereby achieving an untraceable adversarial image for evading DeepFakes attribution. Furthermore, to eliminate imperfection or distortion artifacts and enhance the evasion capability, we implement a hybrid image smoothing operation combining Gaussian Blur and Mean Shift filtering (GBMS) $\mathcal{G}(\cdot)$. The adversarial image is generated as: $x' = \mathcal{T}_{mul}(x|\Phi) = \mathcal{G}(\Phi(x))$.

Guided by the structural prior that model fingerprints are content-coupled modulations, our framework realizes the multiplicative attack principle by parameterizing the adversarial matrix $W$ as a neural function. This design instills an inductive bias that enables effective fingerprint elimination without compromising perceptual fidelity, while overcoming the computational infeasibility and poor generalization inherent in directly optimizing the matrix $W$.

## 5 EXPERIMENTS

We conduct comprehensive experiments to evaluate our attack in evading diverse AMs and defensive mechanisms. Additionally, quantitative analysis in both spatial and frequency domains demonstrates that our method is multiplicative and effectively eliminates fingerprints in generated images.

### 5.1 EXPERIMENTS SETUP

Our experiments span 7 GANs, 5 DMs, and 4 datasets against 6 advanced AMs, and implement 8 attack methods, including 4 transferable attacks (PGD, BIM, MIFGSM, DiffAttack (Chen et al., 2024a)), 3 black-box methods (transformation, FakePolisher (Huang et al., 2020), TraceEvader (Wu et al., 2024)), and 1 regeneration attack Zhao et al. (2024). Besides, we retrain DNA-Det for attributing 3 latest DMs (SD3 (Esser et al., 2024), FLUX (Labs et al., 2025), PixArt (Chen et al., 2024b)) with DiTFake dataset (Li et al., 2025), named **DNA-Det-DMs** for verifying the effectiveness of our method in eliminating fingerprints of DMs. Details in the Appendix A.4.1, Table 3.

### 5.2 EFFECTIVENESS AGAINST ATTRIBUTION METHODS

We evaluate the effectiveness of our proposed method in evading DeepFakes attribution technologies while preserving image quality and demonstrating its universality across diverse GANs and DMs.

*1) Effectiveness:* Our method achieves superior evasion performance against AMs in black-box settings. As shown in Table 1, it attains the highest ASR on four AMs and the second-highest on two others, significantly outperforming existing SOTA methods. Notably, our method achieves an average ASR of 97.08%, substantially higher than TraceEvader's 87.11%. While the regeneration attack achieves considerable ASR in partial AMs, its performance remains suboptimal compared

to our approach. Particularly, its ASR degrades to 39.71% and 0.0% against POSE and LTracer, respectively, highlighting the necessity of fingerprint elimination for attack.

*2) Universality:* Our method effectively eliminates fingerprints from both GAN- and DM-generated DeepFakes, demonstrating broad applicability across diverse GMs. Notably, it achieves 98.89% ASR against LTracer and near-100% ASR on *DNA-Det-DMs*, attacking DeepFakes into real or SD3-generated images, validating its efficacy in eliminating DM-specific fingerprints. More in A.4.2.

*3) Imperceptibility:* Both quantitative and visual evaluations confirm the imperceptibility of our attack. As shown in Table 1, our method matches TraceEvader in image fidelity, achieving 0.963 SSIM and 0.093 LPIPS. Visual comparisons (original vs. adversarial) are provided in Appendix A.4.2, Figure 13, confirming high visual quality preservation. Additionally, confusion matrices in Appendix A.4.2, Figure 11 quantitatively demonstrate that our attack reliably misleads AMs into classifying adversarial DeepFakes as real images, empirically validating perceptual indistinguishability.

## 5.3 Effectiveness Against Defensive Mechanisms

To better assess our attack against the adaptive defense, we identify two distinct scenarios:
*1) Black-Box Scenario:* the defender is unaware of the existence of our attack and can not directly utilize our attack model to generate adversarial images for improving the robustness of AMs. However, the defender can attempt to enhance AMs by creating adversarial images through all other known attacks. *2) White-Box Scenario:* the defender is fully aware of our method and can directly utilize adversarial images created by the proposed method to enhance AMs.

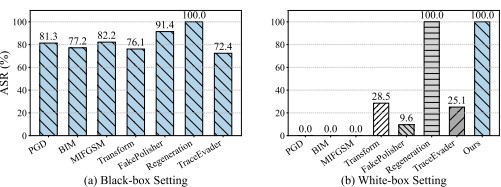

Figure 3: ASR of our method against enhanced AMs. (a) ASR of our attack against AMs enhanced with the augmentation samples indicated on the x-axis. (b) ASR of different attack methods under the white-box setting.

*1) Against Adversarial Training:* Experimental results show neither white-box nor black-box defenses can effectively resist our attacks through adversarial training. As shown in Fig 3, our attack achieves over 72.39% ASR in black-box scenarios. Even when defenders directly use our adversarial images to enhance DNA-Det, no mitigation effect is observed, because the adversarial image contains no information related to the source model, and the ASR even reaches 100%.

*2) Against Approximate Inversion:* We demonstrate that training deep neural networks to recover the original image from the adversarial image is infeasible. The inverted images maintain an ASR of 97.68% and 99.97%, respectively, validating the defensive resistance of our method. This is consistent with the conclusion of our defense difficulty analysis. More details in the Appendix A.4.2.

## 5.4 Ablation Study and Quantitative Analysis

**Ablation Study Analysis:** Experimental results validate the importance of components ($\mathcal{U}_s, \mathcal{U}_t$, GBMS) and function design ($\mathcal{L}_{spatial}, \mathcal{L}_{spectral}$) in our framework. As shown in Table 1, while removing individual components does not degrade image quality, it significantly reduces ASR. For instance, although removing GBMS still achieves SOTA, the average ASR drops from 97.08% to 89.82%. Notably, without $\mathcal{L}_{spectral}$, our

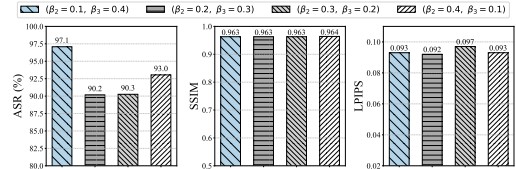

Figure 4: Ablation study on the weighting parameters $\beta_2$ and $\beta_3$ with $\beta_1$ fixed at 0.5, showing their impact on ASR, SSIM, and LPIPS.

method achieves only 50.04% ASR against Reverse AMs. These results conclusively demonstrate the necessity of all designed components and loss functions.

Furthermore, the parameter configuration $(\beta_1, \beta_2, \beta_3)$=(0.5,0.1,0.4) is empirically verified as the optimal choice, achieving the best trade-off between attack effectiveness and perceptual quality. As shown in Figure 4, this setting yields the highest ASR of 97.1%, outperforming all alternative configurations. Crucially, this gain in ASR is achieved without compromising perceptual quality: SSIM and LPIPS remain stable at 0.963–0.964 and 0.092–0.097, respectively.

It is worth noting that our method fundamentally operates by eliminating fingerprints, which inherently requires more modifications than subtle perturbations. While our approach introduces slightly more distortion than TraceEvader, it still achieves perceptual quality superior to many existing methods, striking a favorable balance between attack strength and visual integrity. We will further investigate more efficient fingerprint-elimination mechanisms with minimal distortion in future work.

**Frequency Domain Analysis:** Analysis results indicate that our attack can effectively eliminate the fingerprint left by GANs and DMs. As shown in Figure 1, images attacked by our method exhibit significant differences from the original ones in the frequency domain. This visual evidence indicates that our attack successfully eliminates the source model's fingerprint from DeepFakes.

**Image Domain Analysis:** Distance and correlation analysis both demonstrate that the proposed attack method exhibits multiplicative characteristics rather than additive:

*1) L2 distance among Residual Components:* The residual component $\Delta = \mathcal{T}(x) - x$ verifies attack nature: fixed/stable $\Delta$ indicates additive attacks, while variable $\Delta$ signifies multiplicative behavior. As visualized in Figure 5, our attack exhibits significantly higher variance than TraceEvader's stable residuals, confirming its multiplicative characteristics as the L2 distance mainly within [10,30].

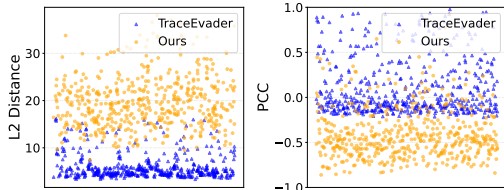

Figure 5: Distribution of L2 distances between $\Delta$ (left) and Pearson Correlation Coefficient (PCC) between $\Delta$ and original images (right). See A.1.2 and Figure 9,10 for more.

*2) Correlation between $\Delta$ and Original Images:* The residual components caused by multiplicative attack $\Delta = x \odot W - x = x \odot (W - 1)$ exhibit higher correlation with original images than those caused by additive attack $\Delta = p$. As shown in Figure 5, the $\Delta$ caused by our method exhibits high correlation with original images, with the absolute value of PCC mainly within [0.5,1]. This strong correlation is a distinctive signature of multiplicative operations, contrasting sharply with TraceEvader's residuals (PCC mainly within [0,0.25]).

**Computational Efficiency Analysis:** Our method achieves high computational efficiency by eliminating fingerprints in a single forward pass, enabling simultaneous evasion of all AMs in a black-box setting. In contrast, transferable adversarial attacks typically require multiple iterative optimization steps and access to the target AMs. Compared to existing black-box approaches, our method achieves substantially higher inference efficiency: it takes only 60.6s to generate 20k adversarial images, significantly faster than the 732.7s required by TraceEvader. (More results in the Table 7.)

## 6 CONCLUSION AND LIMITATION

In this paper, we analyze the principles underlying existing attack methods and reveal a critical structural limitation: current attacks are additive, inherently preserving GMs' fingerprints, making them vulnerable to defense. We argue that true untraceability in DeepFakes can be achieved via multiplicative attacks, and provide theoretical evidence that such attacks are statistically non-invertible, posing a fundamental challenge to existing defensive strategies. Building on this insight, we design a universal attack framework in which the adversarial model learns to eliminate GM fingerprints using only real data, requiring no knowledge of AMs and ensuring broad applicability across diverse GMs. Experiments show our method achieves an ASR of 97.08% across 6 SOTA AMs and 12 GMs, surpassing existing SOTA approaches. It remains highly effective under defense, maintaining an ASR exceeding 72.39%. Our work highlights the emerging threat of multiplicative attacks and underscores the urgent need for more robust attribution mechanisms.

While our method successfully eliminates model fingerprints, it requires more structural modifications than additive attacks. Although visually imperceptible, we aim to optimize efficiency in future work and explore more advanced attack-defense co-evolution strategies.

ACKNOWLEDGMENTS

This research was supported by China National Natural Science Foundation with No. 62441228, Science and Technology Tackling Program of Anhui Province, No.202423k09020016. We also thank Lenovo Research for their support during this research.

ETHICS STATEMENT

This work investigates the elimination of traceable fingerprints in DeepFake generation, a capability that inherently raises ethical concerns. On one hand, techniques for producing untraceable synthetic media could be misused to facilitate disinformation, impersonation, and privacy violations. On the other hand, we argue that rigorously probing the limits of existing attribution technologies is essential for developing more robust and trustworthy countermeasures.

To mitigate potential misuse, we emphasize the following: *1) Intended Purpose:* Our method is developed strictly for research, aiming to expose vulnerabilities in current DeepFake attribution frameworks. *2) Restricted Scope:* All experiments use only publicly available datasets under controlled settings; no real-world personal or sensitive data are involved. *3) Dual-Use Awareness:* While our work demonstrates the feasibility of fingerprint elimination, its primary contribution lies in alerting the community to critical weaknesses in existing attribution technologies. By revealing these limitations, we hope to catalyze research into stronger watermarking, authentication, and regulatory safeguards. *4) Responsible Disclosure:* We provide sufficient implementation details to ensure academic reproducibility, but deliberately omit end-to-end tooling that could be readily weaponized.

We fully acknowledge the dual-use nature of this research and urge the community to interpret our findings not as a blueprint for malicious DeepFake creation, but as a proactive step toward anticipating adversarial capabilities and building more resilient defense mechanisms.

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

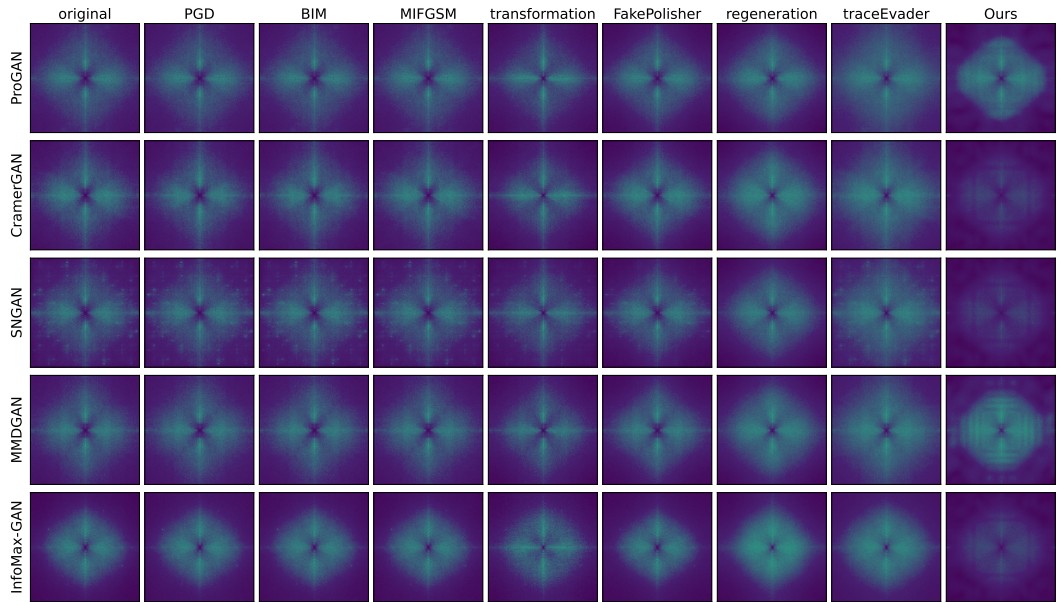

Figure 6: Spectral analysis of high-frequency components (part 1).

# A   APPENDIX

In this appendix, we provide the following supplementary materials:

- A.1 presents a detailed analysis of existing attack methods. Specifically, in A.1.1, we categorize and summarize prior approaches; in A.1.2, we empirically validate their additive nature through multiple analyses, including L2 distance between residual components, correlation between residuals and original images, and frequency-domain characterization, demonstrating their fundamental limitation in effectively eliminating model-specific fingerprints.

- A.2 provides details of the theoretical analysis of the multiplicative attack. Specifically, in A.2.1 we provide a complete proof of Theorem 1, establishing the theoretical existence of the multiplicative adversarial matrix. In A.2.2, we provide a proof for the defense difficulty analysis.

- A.3 details the architectural design of our attack model.

- A.4 elaborates on experimental configurations and presents extended results and discussions. Concretely, A.4.1 specifies the AMs and baseline attacks used in our evaluation, along with implementation details of our framework, defense mechanisms, and inversion model architectures. In A.4.2, we provide an in-depth analysis of our method's effectiveness in evading both state-of-the-art AMs and defensive strategies, demonstrating its capability to eliminate fingerprints from both GANs and DMs while preserving high visual fidelity.

- Finally, we disclose the role of large language models (LLMs) in the preparation of this manuscript.

## A.1   ANALYSIS OF EXISTING ATTACK

### A.1.1   METHOD ANALYSIS

Existing attacks against AMs fundamentally rely on additive perturbations, which can be summarized as follows:

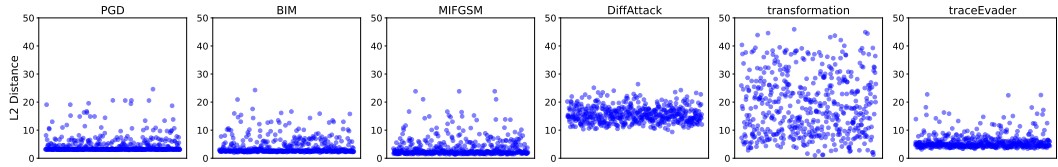

Figure 9: Distribution of L2 distances between residual components.

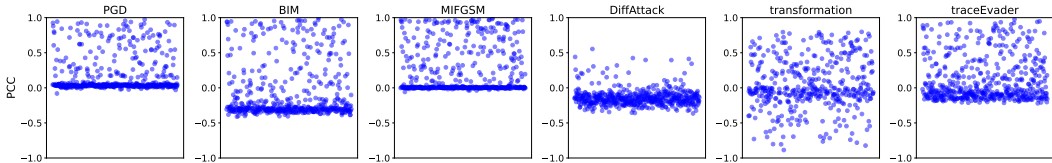

Figure 10: Pearson Correlation Coefficient (PCC) between residual components and the original image.

*1) Transformation-based attack*: Five types of transformation operations: noise, blurring, cropping, JPEG compression, relighting and random combination of them are employed to perturb images (Yu et al., 2019; Yang et al., 2022). Crucially, these transformations either directly add perturbations to the image (e.g., Gaussian noise $\delta$ where $x' = x + \delta$) or can be reformulated as additive operations in transformed domains. For instance, blurring operations apply additive noise in frequency domain, while JPEG compression introduces quantization errors that manifest as additive perturbations in pixel space.

*2) Transferable adversarial example*: The BIM and MI-FGSM are leveraged to generate adversarial examples in a white-box setting against (Wu et al., 2024). Specifically, they solve optimization problems of the form $x' = \arg\max_{x+\delta} \mathcal{L}(x + \delta, y)$ subject to $||\delta||_p \leq \epsilon$, where $\delta$ is the additive perturbation. These perturbations are then transferred to other AMs in black-box settings, maintaining their additive nature.

*3) Black-box and universal attack*: (Wu et al., 2024) adds a universal perturbation learned from DeepFakes to the high-frequency components of images. This perturbation follows the additive model $\Delta x = \delta$, where $\delta$ is a fixed noise pattern added to all images. Similarly, Gaussian blurring and mean shift applied to low-frequency components can be viewed as domain-specific additive perturbations.

All these methods operate within the additive perturbation paradigm, where adversarial examples are generated as $x' = x + \delta$.

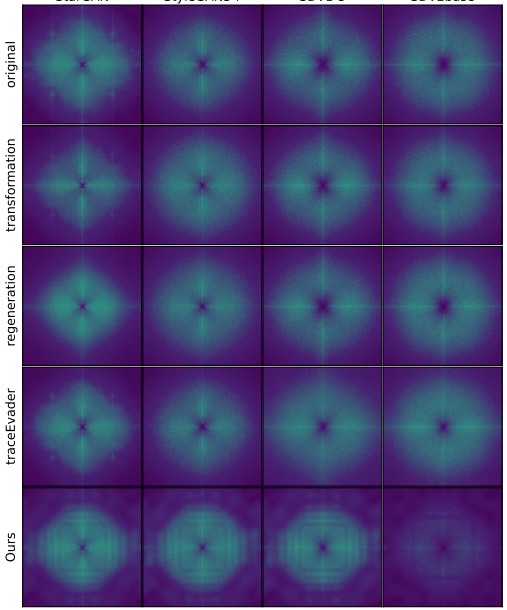

Figure 7: Spectral analysis of high-frequency components (part 2).

### A.1.2 EXPERIMENTAL ANALYSIS

Through analysis of the L2 distance between residual components and the correlation between residuals and original images, we establish that all existing adversarial attacks fundamentally operate within the additive perturbation paradigm. More-

over, through frequency domain analysis of high-frequency components before and after adversarial attacks, we demonstrate that existing attack methods fail to eliminate model fingerprints.

*1) L2 distance among Residual Components:*
Additive attacks generate adversarial examples by directly superimposing a perturbation $p$ onto original images, resulting in $x' = x + p$. Consequently, the residual component $\Delta x = x' - x$ should precisely equal the added perturbation $p$. A defining characteristic of additive attacks is that the L2 distance between residual components across different samples remains relatively stable within a constrained range.

As demonstrated in Figure 9, all existing attack methods exhibit this stability pattern. Specifically, the L2 distances of residual components for PGD, BIM, MI-FGSM, and TraceEvader consistently concentrate within the narrow interval [0, 10]. DiffAttack shows slightly higher values within [10, 20], while transformation-

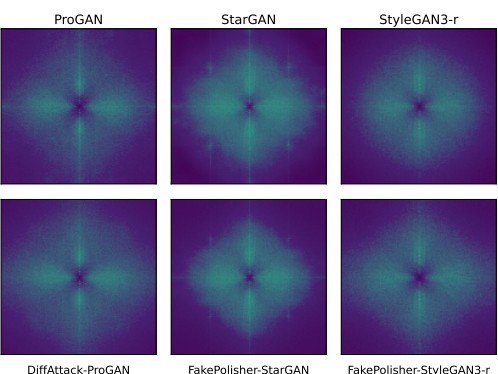

Figure 8: Spectral analysis of high-frequency components (part 3).

based attacks, which incorporate multiple operations, display a broader distribution. Nevertheless, most of the residuals still fall within the [0, 20] range, maintaining the characteristic stability of additive perturbations.

This consistent concentration of L2 distances provides empirical evidence that these attacks adhere to the additive perturbation model $x' = x + p$, where the residual $\Delta x$ directly corresponds to the added perturbation $p$.

*2) Correlation between $\Delta$ and Original Images:* In the additive attack framework, the residual component $\Delta x = p$ should exhibit minimal correlation with the original image $x$. To quantify this relationship, we employ the Pearson Correlation Coefficient (PCC), where values approaching 0 indicate no linear correlation, while absolute values closer to 1 indicate stronger correlation.

Figure 10 reveals that residuals generated by existing attack methods consistently demonstrate low correlation with original images. Specifically, PGD and MI-FGSM produce residuals with PCC values approaching zero, while DiffAttack and TraceEvader yield slightly higher but still minimal correlations ( absolute value of PCC within [0,0.25] ).

This consistently low correlation pattern across diverse attack methods provides compelling evidence that existing approaches operate within the additive perturbation paradigm.

*3) Frequency Analysis:* Model fingerprints exhibit pronounced characteristics in the frequency domain, manifesting as distinct patterns (Frank et al., 2020). Our comprehensive evaluation spans 9 GMs, including 7 GAN variants and 2 DMs. As shown in Figures 6, 7 and 8, the high-frequency spectra of adversarial examples generated by all existing methods, including additive attacks, reconstruction-based approaches (FakePolisher) and regeneration attack maintain remarkable similarity to those of original DeepFakes.

These empirical findings collectively demonstrate that existing adversarial attack methods fail to effectively eliminate model fingerprints, which are critical for DeepFakes attribution. This finding provides critical insight into the limitations of current attribution attack methods against AMs and motivates our proposed multiplicative attack approach.

## A.2 DETAILS OF THEORETICAL ANALYSIS OF MULTIPLICATIVE ATTACK

### A.2.1 EXISTENCE OF ADVERSARIAL MATRIX $W$

We define the adversarial attack optimization problem as follows:
$$W^* = \arg \max_W \|\mathcal{F}(x \odot W) - \mathcal{F}(x)\| \quad \text{s.t.} \quad \|\Delta x\|_2 \leq \Delta,$$
where $\Delta x = x \odot (W - I)$ is the residual component between the adversarial image and the original image, and $\| \cdot \|_2$ is the L2 norm. Assuming $\mathcal{F}(x)$ is differentiable at $x$, the first-order Taylor

expansion around $x$ is:

$$\mathcal{F}(x \odot W) \approx \mathcal{F}(x) + J(x) \cdot \Delta x,$$

where $J(x) \in \mathbb{R}^{m \times n}$ is the Jacobian matrix:

$$J(x)_{ij} = \frac{\partial \mathcal{F}_i(x)}{\partial x_j}.$$

The optimization problem can be rewritten as:

$$W^* = \arg \max_W \|J(x) \cdot \Delta x\| \quad \text{s.t.} \quad \|\Delta x\|_2 \leq \Delta.$$

We now prove the existence of an adversarial matrix $W$ satisfying this goal.

**Theorem 1** (Existence of Multiplicative Adversarial Matrix). *Let $F(x)$ be differentiable in a neighborhood of $x$, and let the Jacobian matrix $J(x) \in \mathbb{R}^{m \times n}$ be locally Lipschitz around $x$. If the input $x \in \mathbb{R}^n$ satisfies the non-degeneracy condition: There exists at least one coordinate $j \in \{1, 2, \ldots, n\}$ such that $x_j \neq 0$ and $\frac{\partial F_i(x)}{\partial x_j} \neq 0$ for some $i$, equivalently, the restricted spectral norm $\gamma = \max_{\|v\|_2=1, \, v \in S} \|J(x) v\|_2 > 0$, then there exists a multiplicative perturbation matrix $W$ such that for any $\epsilon \in (0, \gamma \cdot \Delta]$,*

$$\|J(x) \cdot \Delta x\|_2 \geq \epsilon \quad \text{and} \quad \|\Delta x\|_2 \leq \Delta,$$

*where we set $\Delta x := x \odot (W - \mathbf{1})$ and $S = \{ v \in \mathbb{R}^n \mid v_j = 0 \text{ whenever } x_j = 0 \}$ is the admissible perturbation subspace, and the restricted Jacobian $J_S(x)$ has spectral norm $\gamma > 0$ as defined above. Moreover, choosing $\Delta$ small enough and constructing $W$ by $W_j = 1 + \Delta x_j/x_j$ for $x_j \neq 0$ (and $W_j = 1$ otherwise) keeps $x \odot W$ in the valid pixel range, and the first–order Taylor error is bounded by $O(\Delta^2)$.*

*Proof.* We proceed in five steps to construct $W$ and validate the inequality.

1. Structural Constraints on $\Delta x$: Define $\Delta x = x \odot (W - 1)$. For components where $x_j = 0$, we have $\Delta x_j = 0$. Thus, $\Delta x$ is restricted to the subspace $S = \{v \in \mathbb{R}^n \mid v_j = 0 \text{ if } x_j = 0\}$.

2. Admissible Perturbation Directions: By the non-degeneracy condition, the projection $J_S(x)$ (retaining only columns of $J(x)$ where $x_j \neq 0$) satisfies $\|J_S(x)\|_F > 0$. Hence, there exists a non-zero direction $v \in S$ such that $\|J(x) \cdot v\| > 0$.

3. Optimal Perturbation via SVD: Let $J_S(x) = U \Sigma V^\top$ be the SVD of $J_S(x)$, where $\sigma_1 = \|J_S(x)\|_2$ is the largest singular value and $v_1 \in S$ is the corresponding right singular vector. Define:

$$\Delta x = \Delta \cdot v_1.$$

Then $\|\Delta x\|_2 = \Delta$, and:

$$\|J(x) \cdot \Delta x\| = \Delta \cdot \|J_S(x) \cdot v_1\| = \Delta \cdot \sigma_1.$$

Set $\gamma = \sigma_1$, so $\|J(x) \cdot \Delta x\| = \gamma \cdot \Delta$.

4. Constructing $W$: From $\Delta x = x \odot (W - I)$, solve for $W$:

$$W_j = \begin{cases} 1 + \frac{\Delta x_j}{x_j}, & x_j \neq 0, \\ 1, & x_j = 0. \end{cases}$$

This ensures $\Delta x = x \odot (W - I)$ and $\|\Delta x\|_2 \leq \Delta$.

5. Validating Arbitrary $\epsilon \in (0, \gamma \cdot \Delta]$: For any $\epsilon \in (0, \gamma \cdot \Delta]$, let $\delta = \epsilon/\gamma \leq \Delta$. Define:

$$\Delta x' = \delta \cdot v_1.$$

Then $\|\Delta x'\|_2 = \delta \leq \Delta$, and:

$$\|J(x) \cdot \Delta x'\| = \gamma \cdot \delta = \epsilon.$$

Construct $W'$ using the same formula as above.

If $J_S(x) \neq 0$ (i.e., $\gamma > 0$), there exists a perturbation $\Delta x \in S$ with $\|\Delta x\|_2 \leq \Delta$ such that $\|J(x) \cdot \Delta x\| \geq \epsilon$ for all $\epsilon \in (0, \gamma \cdot \Delta]$. $\qquad \square$

A.2.2 PROOFS FOR DEFENSE DIFFICULTY ANALYSIS

**Proposition 1** (Non-identifiability without paired supervision). *Assume $x' = P(x \odot W) + \eta$, where $P$ is a deterministic pre-processing operator and $\eta$ is measurement noise. In the idealized case $P = I$, $\eta = 0$, the decomposition of $x'$ into $(x, W)$ is not identifiable: there exist infinitely many $(\tilde{x}, \tilde{W})$ such that $\tilde{x} \odot \tilde{W} = x'$.*

*Proof.* Let $S = \{j : x_j \neq 0\}$. For any $\delta \in \mathbb{R}^n$ with $\|\delta\|_\infty < 1$ and $\delta_j = 0$ for $j \notin S$, define $\tilde{x} = x \odot (1 + \delta)$ and $\tilde{W} = W \odot (1 + \delta)^{-1}$. Then $\tilde{x} \odot \tilde{W} = x \odot (1 + \delta) \odot W \odot (1 + \delta)^{-1} = x \odot W = x'$. Varying $\delta$ yields infinitely many valid decompositions, hence non-identifiability. This non-identifiability persists under any deterministic $P$, because if $x_1 \odot W_1 = x_2 \odot W_2$, then $P(x_1 \odot W_1) = P(x_2 \odot W_2)$. The presence of noise $\eta$ only exacerbates the ambiguity. $\square$

**Lemma 1** (Pre-processing does not fix identifiability). *If $P$ is deterministic and injective on the image range of interest, then non-identifiability of $(x, W)$ from $x'$ under $P = I, \eta = 0$ implies non-identifiability under $x' = P(x \odot W)$ as well.*

*Proof.* If $x_1 \odot W_1 = x_2 \odot W_2$, then $P(x_1 \odot W_1) = P(x_2 \odot W_2)$ by determinism. Injectivity is only needed if one tries to recover $x \odot W$ from $x'$; it does not help distinguish different $(x, W)$ pairs producing the same product before $P$. $\square$

**Proposition 2** (CRLB-style lower bound with paired supervision). *Consider the per-pixel model $x'_j = x_j W_j + \eta_j$, where $\eta_j \sim \mathcal{N}(0, \sigma^2)$ i.i.d. represents measurement noise (e.g., from image acquisition or pre-processing). Given $N$ i.i.d. pairs $(x^{(k)}, x'^{(k)})$, any unbiased estimator $\widehat{W}_j$,*

$$\mathrm{Var}(\widehat{W}_j) \geq \frac{\sigma^2}{N \, \mathbb{E}\left[(x_j^{(k)})^2\right]}.$$

*Consequently, to achieve $\mathbb{E}(\widehat{W}_j - W_j)^2 \leq \varepsilon^2$ one needs $N \gtrsim \sigma^2 / (\varepsilon^2 \, \mathbb{E}[(x_j^{(k)})^2])$.*

*Sketch.* The log-likelihood of $\{x_j'^{(k)}\}_{k=1}^N$ given $W_j$ is Gaussian with mean $x_j^{(k)} W_j$ and variance $\sigma^2$. The Fisher information is $I_j(W_j) = \frac{1}{\sigma^2} \sum_{k=1}^N (x_j^{(k)})^2$. The Cramér–Rao lower bound yields $\mathrm{Var}(\widehat{W}_j) \geq I_j(W_j)^{-1}$; taking expectation over the data distribution gives the stated bound. $\square$

**Remark 1** (Effect of standard pre-processing). *If $P$ is $L$-Lipschitz and $F$ is locally Lipschitz, then for a feasible $\Delta x = x \odot (W - \mathbf{1})$ with $\|\Delta x\|_2 \leq \Delta$, $\|F(P(x \odot W)) - F(P(x))\|_2 \leq L \|\Delta x\|_2 + \mathcal{O}(\Delta^2)$. Thus $P$ changes constants but not the identifiability conclusion of Proposition 1.*

A.3 THE DETAILS OF THE MODEL ARCHITECTURE

Table 2: Proposed adversarial model architecture. $H$ and $W$ denote input image height and width; InstNorm = Instance Normalization; $\times 5$ = repeated 5 times; Dual = two sequential convolutional operations per block.

| Component | Layer | Kernel | Stride | Padding | Output Shape | Activation/Normalization |
|---|---|---|---|---|---|---|
| **Encoder** | ConvLayer (Input) | $9 \times 9$ | 1 | Reflect | $32 \times H \times W$ | ReLU + InstNorm |
| | ConvLayer (Downsample) | $3 \times 3$ | 2 | Reflect | $64 \times \frac{H}{2} \times \frac{W}{2}$ | ReLU + InstNorm |
| | ConvLayer (Downsample) | $3 \times 3$ | 2 | Reflect | $128 \times \frac{H}{4} \times \frac{W}{4}$ | ReLU + InstNorm |
| **Residual Blocks** | ResidualLayer $\times 5$ | $3 \times 3$ (dual) | 1 | Reflect | $128 \times \frac{H}{4} \times \frac{W}{4}$ | Layer 1: ReLU + InstNorm; Layer 2: Linear + InstNorm; Skip connection |
| **Decoder** | DeconvLayer (Upsample 1) | $3 \times 3$ | 1 | Reflect | $64 \times \frac{H}{2} \times \frac{W}{2}$ | ReLU + InstNorm |
| | DeconvLayer (Upsample 2) | $3 \times 3$ | 1 | Reflect | $32 \times H \times W$ | ReLU + InstNorm |
| | ConvLayer (Output) | $9 \times 9$ | 1 | Reflect | $3 \times H \times W$ | Linear |

As systematically documented in Table 2, our adversarial model comprises an encoder-decoder architecture.

Table 3: Summary of Attribution Models, Generative Models, and those used in the experiments. SDv1-5 and SDv2-base respectively, indicate Stable Diffusion v1-5 and Stable Diffusion v2-base. Besides, we retrain DNA-Det to attributing (SD3 (Esser et al., 2024), FLUX (Labs et al., 2025), PixArt (Chen et al., 2024b)) with DiTFake dataset (Li et al., 2025).

| | ProGAN | SNGAM | MMDGAN | CramerGAN | InfoMaxGAN | StarGAN | StyleGAN3 | SDv1-5 | SDv2base |
|---|---|---|---|---|---|---|---|---|---|
| DNA-Det | ✓ | ✓ | ✓ | | ✓ | | | | |
| AttNet | ✓ | ✓ | ✓ | ✓ | | | | | |
| DCT | ✓ | ✓ | ✓ | ✓ | | | | | |
| Reverse | ✓ | ✓ | ✓ | ✓ | | | | | |
| POSE | ✓ | | | | | ✓ | ✓ | | |
| LTracer | | | | | | | | ✓ | ✓ |

The encoder processes the input image through a cascade of convolutional and residual layers to extract features. It comprises: 1) Three initial convolutional layers: The first layer employs a 9×9 kernel with stride 1, transforming the 3-channel input into 32 feature maps. The subsequent two layers use 3×3 kernels with stride 2 for spatial downsampling, progressively expanding the channel depth to 64 and then 128 while halving the spatial resolution at each step. All convolutional layers utilize reflection padding, instance normalization, and ReLU activation. 2) Five residual refinement blocks: Each block processes the 128-channel features through two sequential 3×3 convolutional layers with stride 1. The first layer in each block applies ReLU activation, while the second operates linearly (without activation) to preserve gradient flow. Skip connections sum the block input with the processed features, mitigating degradation in deep networks and preserving details. Instance normalization is applied after each convolutional operation to ensure consistent feature distributions.

The decoder reconstructs the high-resolution output via two upsampling stages and an output synthesis layer: 1)Upsampling stages: Stage 1: Upsamples the 128-channel encoder output to 64 channels, followed by a 3×3 convolutional layer, instance normalization, and ReLU activation. Stage 2: Further upsamples to 32 channels through an identical operation sequence (interpolation, 3×3 convolution, normalization, ReLU). 2) Output synthesis layer: A final 9×9 convolutional layer with stride 1 and linear activation maps the 32-channel features to the 3-channel output image.

## A.4 The Details of the Experiments

### A.4.1 Experimental Setup Details

This section comprehensively details the experimental framework employed in our evaluation, including the victim models under attack, benchmark adversarial methodologies, and implementation details utilized for validation. We systematically describe the implementation specifics of both adversarial training and approximate inversion techniques (Defense technology).

*1) Victim Model (AMs):* We conduct experiments against 6 advanced attribution technologies, including DNA-Det (Yang et al., 2022), AttNet (Yu et al., 2019), DCT (Frank et al., 2020), Reverse (Asnani et al., 2023), POSE (Yang et al., 2023), and LTracer Wang et al. (2024). As summarized in Table 3, our experiments span 7 GANs, 5 DMs and 4 datasets. All GANs employed in our evaluation were pre-trained on established facial datasets. Specifically, StyleGAN was trained on the high-fidelity FFHQ dataset, while alternative GAN architectures were trained on the CelebA dataset. The two DMs were trained on the large-scale LAION dataset. We leveraged these pre-trained generative models to generate DeepFakes for comprehensive experimental evaluation. And for all AMs, we follow their original default Settings.

In addition, we implement DNA-Det for attributing 3 latest DMs (SD3 (Esser et al., 2024), FLUX (Labs et al., 2025), PixArt (Chen et al., 2024b)) with DiTFake dataset (Li et al., 2025) by ourselves, which is used to verify the effectiveness of our method in eliminating fingerprints of diffusion models. We used DiTFake dataset to train it into a four-class classification model (real, SD3, FLUX, PixArt), 5000 images per class, 20,000 images in total) to train DNA-Det. We used 16,000 images for model training and 2000 for validation. 2000 for testing and attacking. We adopted the same default training settings as for the CelebA dataset. The attribution accuracy of the trained model is 96.65%.

Table 4: Performance of attribution models on clean (non-adversarial) samples

|          | DNA-Det | AttNet | DCT   | Reverse | POSE  | LTracer |
|----------|---------|--------|-------|---------|-------|---------|
| Accuracy | 100.0   | 99.43  | 99.07 | 99.66   | 85.43 | 98.89   |

*2) Baseline (Attack methods):* For comparative analysis, we implement 8 attack methods, including 4 transferable attacks (PGD, BIM, MIFGSM, DiffAttack (Chen et al., 2024a)), and 3 black-box methods, including transformation-based attack, FakePolisher (Huang et al., 2020) and TraceEvader (Wu et al., 2024). Besides, we adopt 1 regeneration attack for comparison Zhao et al. (2024). For transferable methods, adversarial images are created by attacking the DNA-Det in a white-box setting and transferred against other AMs aligned with (Wu et al., 2024). Due to the high computational overhead of DiffAttack, we limited its application to generating adversarial examples specifically against ProGAN-generated DeepFakes. Similarly, FakePolisher's implementation is constrained to facial imagery through its domain-specific dictionary construction; consequently, we evaluated this attack exclusively on facial DeepFakes while excluding LTracer due to dataset domain mismatch. For regeneration attacks, we employed Stable Diffusion-v2.1 as the reconstruction model.

*3) Implementation Details :* Our attack model is trained on the CelebA dataset to eliminate fingerprints in face images and on the ImageNet test set to remove fingerprints from DMs. We employ the Adam optimizer with an initial learning rate of 1e-4 and use a cosine annealing strategy to adjust the learning rate. The weights in $L_{total}$ $\beta_1, \beta_2, \beta_3$ are set as $\{0.5, 0.1, 0.4\}$.

*4) Adversarial Training Details:* We implemented adversarial training to enhance the robustness of the DNA-Det detector against our proposed adversarial attack. Specifically, we preserved the original architecture and training configuration of DNA-Det while replacing 50% of the training samples with adversarial images generated by our method, maintaining the ground-truth labels throughout the training process. After completing the adversarial training phase, we evaluated the attack success rate of our method against enhanced AMs.

*5) Approximate Inversion Details:* To investigate the potential of approximate inversion as a defense mechanism against our adversarial attacks, we implemented two inversion models based on established denoising architectures: DnCNN and an Autoencoder. Both models were trained to reconstruct clean images from adversarial examples, with the adversarial inputs serving as the source domain and the corresponding clean images as ground-truth targets. The training objective employed the Mean Squared Error loss function. After model convergence, we evaluated the effectiveness of this defense strategy by processing the reconstructed images through the DNA-Det.

The architectural details of the two networks are as follows: *1) DnCNN:* The DnCNN architecture employed in our experiments consists of a 20-layer deep convolutional layers. The network begins with an initial convolutional layer that transforms the 3-channel input into 64 feature maps using a 3×3 kernel with stride 1 and padding 1, followed by a ReLU activation function. This is succeeded by 18 intermediate layers, each comprising a 3×3 convolutional operation maintaining 64 feature channels, batch normalization, and ReLU activation. The final layer projects the feature representation back to the original 3-channel output space through another 3×3 convolution, with a sigmoid activation ensuring the output values remain within the valid [0,1] range. *2) Autoencoder:* The Autoencoder architecture implements a symmetric encoder-decoder structure. The encoder pathway begins with a convolutional layer that processes the 3-channel input into 64 feature maps using a 4×4 kernel with stride 2 and padding 1, followed by batch normalization and ReLU activation. Three additional convolutional layers progressively increase the channel depth to 128 and 256 while reducing spatial dimensions, culminating in a bottleneck layer that compresses the representation to 256 channels at 8×8 spatial resolution, forming the latent space representation. The decoder reverses this compression process through four transposed convolutional layers that progressively upsample the feature maps while reducing channel depth , each followed by batch normalization and ReLU activation (with the exception of the final layer which employs a sigmoid activation to constrain output values to the [0,1] range). All convolutional weights are initialized using Kaiming normal initialization to ensure stable training dynamics, while batch normalization parameters follow standard initialization protocols. We summarize the architecture of both models in the table 5 and 6.

Table 5: DnCNN Architecture

| Component | Layer Type | Kernel Size | Stride | Padding | Output Shape | Activation/Normalization |
|---|---|---|---|---|---|---|
| Input Layer | Conv2d | $3 \times 3$ | 1 | Reflect | $64 \times H \times W$ | ReLU |
| Intermediate | Conv2d $\times 18$ | $3 \times 3$ | 1 | Reflect | $64 \times H \times W$ | ReLU + BatchNorm |
| Output Layer | Conv2d | $3 \times 3$ | 1 | Reflect | $3 \times H \times W$ | Sigmoid |

Table 6: Autoencoder Architecture

| Component | Layer Type | Kernel Size | Stride | Padding | Output Shape | Activation/Normalization |
|---|---|---|---|---|---|---|
| **Encoder** | Conv2d (Downsample 1) | $4 \times 4$ | 2 | Reflect | $64 \times \frac{H}{2} \times \frac{W}{2}$ | ReLU + BatchNorm |
| | Conv2d (Downsample 2) | $4 \times 4$ | 2 | Reflect | $128 \times \frac{H}{4} \times \frac{W}{4}$ | ReLU + BatchNorm |
| | Conv2d (Downsample 3) | $4 \times 4$ | 2 | Reflect | $256 \times \frac{H}{8} \times \frac{W}{8}$ | ReLU + BatchNorm |
| | Bottleneck | $4 \times 4$ | 2 | Reflect | $256 \times \frac{H}{16} \times \frac{W}{16}$ | ReLU + BatchNorm |
| **Decoder** | ConvTranspose2d (Upsample 1) | $4 \times 4$ | 2 | Reflect | $256 \times \frac{H}{8} \times \frac{W}{8}$ | ReLU + BatchNorm |
| | ConvTranspose2d (Upsample 2) | $4 \times 4$ | 2 | Reflect | $128 \times \frac{H}{4} \times \frac{W}{4}$ | ReLU + BatchNorm |
| | ConvTranspose2d (Upsample 3) | $4 \times 4$ | 2 | Reflect | $64 \times \frac{H}{2} \times \frac{W}{2}$ | ReLU + BatchNorm |
| | Output Layer | $4 \times 4$ | 2 | Reflect | $3 \times H \times W$ | Sigmoid |

### A.4.2 EXPERIMENTAL EVALUATION DETAILS

This section presents more empirical analysis and visual evidence to further substantiate our methodology's effectiveness.

*1) Effectiveness Against Attributions:* Figure 11 presents a comprehensive confusion matrix analysis of AMs under our adversarial attack methodology. The visualization reveals that our attack successfully compromises the DeepFakes attribution capability of all evaluated AMs, significantly degrading their ability to determine the generative origin of DeepFakes. Among them, DCT misclassifies 100% of adversarial images as CramerGAN-generated content, therefore, we exclude CramerGAN-generated DeepFakes attribution from our attack success rate calculations.

Notably, our attack induces AMs to misclassify a substantial proportion of adversarial images as real images, demonstrating the perceptual indistinguishability of our adversarial images from genuine imagery. Most strikingly, when targeting AttNet, our method achieves a perfect 100% misattribution rate to real images, demonstrating that the adversarial images generated by our approach are perceptually indistinguishable from real images. This complete source obfuscation represents a significant advancement over prior art, as it effectively eliminates the distinguishing features that attribution models typically exploit for detection.

*2) Effectiveness Against Defensive Mechanisms:* Figure 12 presents a confusion matrix analysis of DNA-Det's robustness against our adversarial methodology under two prominent defense paradigms: adversarial training and approximate inversion. The visualization reveals that adversarially trained DNA-Det fails to correctly attribute adversarial images, exhibiting degraded performance compared to the baseline model. Specifically, the adversarially trained DNA-Det misclassifies 100% of adversarial images as authentic content, representing a complete collapse of source attribution capability. Similarly, approximate inversion defenses prove ineffective against

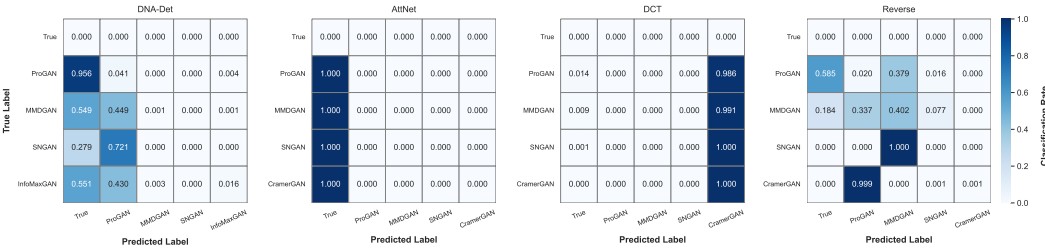

Figure 11: Attribution Performance Analysis: Confusion matrices of the DeepFakes model under the proposed adversarial attack.

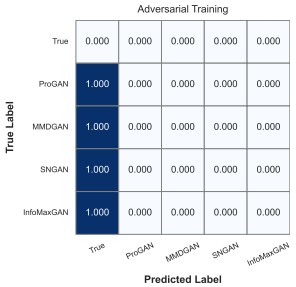
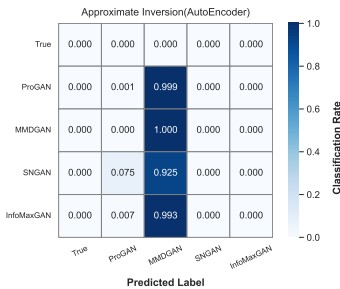

Figure 12: Attribution Performance Analysis: Confusion matrices of DNA-Det (enhanced) under the proposed adversarial attack.

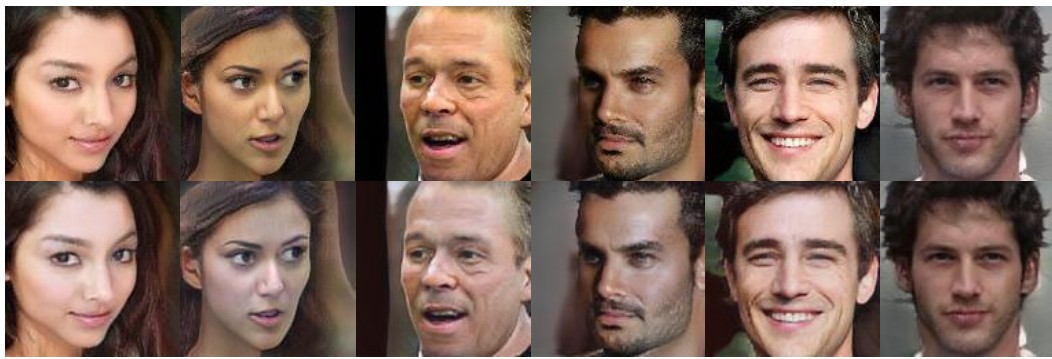

Figure 13: Visualization of the original DeepFakes (top) and adversarial images (bottom) generated by the out attack.

our methodology. As evidenced by the confusion patterns across both inversion approaches, both DnCNN-based and AutoEncoder-based reconstruction models fail to recover traceable fingerprints for accurate attribution. These findings collectively demonstrate the resilience of our attack against state-of-the-art defense mechanisms.

The above experimental results show that: 1) On the one hand, even though the adversarial training defense measures have seen the adversarial images attacked by our method during the training process, they cannot align the correct attribution, which indicates that the multiplicative attack proposed by us effectively eliminates the fingerprints inside DeepFakes, so that AMs cannot effectively perform attribution even if it is enhanced by defense measures. Because there is no information related to the source model in the image after the attack. 2) On the other hand, the defense method based on approximate inversion is also unable to effectively defend, which is consistent with the conclusion of our defense difficulty analysis above. Even if the defender can collect image pairs to train the inversion model, it cannot defend against our attack, because restoring images with the defense of neural networks will introduce new network fingerprints into new images, further reducing the possibility of attribution.

*3) Effectiveness Preserving Image Fidelity:* Figure 13 presents a visual comparison between original DeepFakes and their adversarial counterparts generated by our methodology. The side-by-side examples demonstrate exceptional visual fidelity preservation, with structural similarity (SSIM) scores averaging 0.963 and Learned Perceptual Image Patch Similarity (LPIPS) of 0.093 across the evaluation images. While minor color variations are observable in some regions, the semantic content remains fully preserved, including facial identity, expression, and structural features. This high degree of perceptual consistency confirms that our attack operates within imperceptibility thresholds while effectively compromising attribution model performance, representing a critical balance between attack efficacy and visual Fidelity.

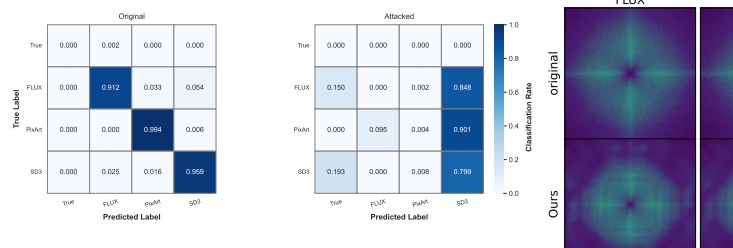

Figure 14: Attribution Performance Analysis: Confusion matrices of the DNA-Det-DMs classifier on clean (left) and attacked (right) DeepFakes generated by FLUX, PixArt, and SD3.

Figure 15: Spectral analysis of high-frequency components in DeepFakes generated by FLUX, PixArt, and SD3 before (top row) and after (bottom row) our attack.

*4) Effectiveness Eliminating DMs' Fingerprints:* We evaluate the effectiveness of our attack method in eliminating DMs' fingerprints across five state-of-the-art diffusion models: Stable Diffusion v1.5, Stable Diffusion v2.1, FLUX, PixArt, and Stable Diffusion 3.

We assess the attack performance on a custom-trained attribution model (DNA-Det-DMs) that was trained to identify DeepFakes generated by the three most recent diffusion models: FLUX, PixArt, and SD3. This model achieves an attribution accuracy of 96.65% on clean DeepFakes, demonstrating its strong discriminative capability under benign conditions. In contrast, our adversarial attack achieves an ASR of nearly 100%, indicating near-total evasion of model attribution.

To visualize the impact of our attack, we present confusion matrices in Figure 14, comparing the DNA-Det-DMs's performance before and after the attack. The left subfigure shows that, on clean DeepFakes, the model accurately distinguishes among images generated by different DMs. However, the right subfigure reveals a dramatic degradation in attribution performance under attack: our adversarial DeepFakes successfully mislead the DNA-Det-DMs to predominantly misattribute all DMs-generated images as either "real" or as originating from SD3, effectively collapsing the model's discriminative power and undermining its forensic utility.

Frequency-domain analysis further validates that our method effectively eliminates the intrinsic fingerprints of DMs, even for the most recent architectures. As illustrated in Figure 7, DeepFakes generated by SDv1.5 and SDv2base after being attacked by our method exhibit significant spectral deviations from their clean counterparts. In Figure 15, we extend this analysis to the three latest DMs (FLUX, PixArt, and SD3), comparing the frequency-domain characteristics of their generated images before and after attack. Consistent with the observations in Figure 7, the adversarial DeepFakes demonstrate substantial spectral difference compared to their original, clean versions.

Collectively, these results demonstrate that our attack successfully suppresses and, in many cases, effectively eliminates the model-specific fingerprints in DM-generated DeepFakes. This holds true even for the latest generation of DMs, demonstrating the broad applicability of our method, even against the latest DMs.

We do not include a dedicated evaluation of typical post-processing pipelines (e.g., JPEG re-compression, resizing, cropping) for the following reasons: 1) Our multiplicative attack is designed to eliminate generator-specific fingerprints within DeepFakes. Once the original fingerprint has been eliminated, subsequent image-level transforms are information-reducing and cannot reconstruct the removed statistics; in practice, they further destroy any residual cues used by attribution models, thus making correct attribution even less likely. 2) TraceEvader (Wu et al., 2024) has already evaluated and reported that operations like JPEG re-compression, resizing, and cropping do not degrade the attack performance; they introduce even greater perturbations to images, which can further increase the attack success rate.

Table 7: Computational efficiency comparison of attack methods. Reported values are end-to-end inference times (in seconds) for generating 20,000 adversarial images on a fixed hardware platform. Lower is better. The best results are marked in **bold**.

| Mehtod | PGD | BIM | MIFGSM | FakePolisher | Regeneration | TraceEvader | Ours |
|---|---|---|---|---|---|---|---|
| Time Cost | 102.62 | 79.54 | 127.62 | 2611.1 | 2180.49 | 732.70 | **60.64** |

## A.5 USE OF LARGE LANGUAGE MODELS (LLMS)

We employed large language models (Qwen) as writing aids to improve the clarity and fluency of certain sections. These tools were used exclusively for language refinement and structural suggestions; they did not contribute to the design of our attack framework, theoretical analysis, or experimental evaluation. All technical content, including algorithms, proofs, and results, is the original work of the authors. We assume full responsibility for the integrity of this manuscript.

