# OpenReview forum: "Untraceable DeepFakes via Traceable Fingerprint Elimination"
_ICLR.cc/2026/Conference — ICLR 2026 Poster_

### Official Review · Reviewer_Spzx · 2025-10-28

**Soundness:** 2
**Presentation:** 3
**Contribution:** 2
**Rating:** 2
**Confidence:** 4

**Summary:**

This paper proposes a "multiplicative attack" framework aimed at generating untraceable deepfakes that evade forensic attribution models. The core idea is to apply pixel-wise multiplicative perturbations to the generated image to suppress identifiable model fingerprints while preserving visual fidelity. The authors present a theoretical justification suggesting that such perturbations make attribution statistically difficult, and propose an optimization-based implementation that enforces perceptual similarity through the combination of multiple loss terms. Extensive experiments are conducted across multiple models, showing that the proposed method significantly reduces attribution accuracy compared to existing attacks while maintaining high image quality, especially under adversarial training.

**Strengths:**

- The paper studies an important and timely problem, i.e., untraceable deepfakes.
- The experiments are comprehensive to a large extent and cover multiple generative and attribution models.
- The paper is well written and easy to follow.
- Code is provided, which should be encouraged. Although in the README the paper states the code is "submitted to AAAI".

**Weaknesses:**

My main concerns focus on the paper’s theoretical claims about the "multiplicative attack" and the supporting propositions. Currently, the theoretical claims made in this paper do not actually support the strong claims regarding multiplicative attacks being uniquely powerful.

- The paper presents its attack as multiplicative, $x' = x \odot W$, but then immediately defines the adversarial residual as $\Delta x = x \odot (W - 1)$ and constrains $|\Delta x|_2 \le \Delta$. This is algebraically identical to writing $x' = x + p(x)$ with $p(x) = x \odot (W-1)$. In other words, the "multiplicative" formulation is just an input-dependent additive perturbation in disguise. Because all of the theorems and propositions are stated and proved in terms of $\Delta x$ (the residual) and its $\ell_2$ geometry, the results apply equally well to additive attacks $x' = x + p$. Therefore, the paper gives no theoretical justification for the claim that multiplicative attacks are somehow fundamentally superior to additive attacks. In fact, since the theorems themselves are modeled under an $\ell_2$ norm constraint, they appear to fit additive models better than the proposed multiplicative model. At best, the paper demonstrates properties of $x + p(x)$ models, not anything special about strict multiplicative structure.

- Theorem 1 is essentially: assume $F$ is differentiable at $x$ and the Jacobian $J(x)$ has a non-zero direction, then there exists a small $\Delta x$ (with $|\Delta x|_2 \le \Delta$) that changes $F(x)$. I have two concerns regarding this theorem.
  - The assumptions already imply the conclusion. Requiring a non-degenerate $J(x)$ is equivalent to assuming the model is sensitive to some input direction, so the "existence" of a perturbation that changes the output is tautological. This is hardly a new theoretical insight, but rather a trivial existence.
  - The authors provide no bound on how large the change in the attribution decision will be, nor any probabilistic results about success across AMs. Changing $F(x)$ in the sense of its vector-valued output does not imply flipping the attribution label, nor does it imply removal of a fingerprint. Thus, it is not guaranteed that the attribution decision will be changed and thus the fingerprint removed.

- Proposition 1 and 2 are also problematic and not specific to multiplicative models. The paper argues that multiplicative attacks are "statistically non-invertible" and supports this with a per-pixel linear-Gaussian model $x'_j = x_j W_j + \eta_j$ and a Cramér–Rao lower bound $\mathrm{Var}(\hat W_j) \ge \sigma^2/(N \mathbb{E}[x_j^2])$. This derivation is problematic for multiple reasons.
  - The noise $\eta$ seems to be artificially introduced. In the attack model the adversary deterministically crafts $x'$; there is no physical measurement noise generated by the attacker. The paper only introduces $\eta$ later so that standard CRLB formulas can be applied. It is unclear why such noise is present and why it would follow the Gaussian distribution. More confusingly, why is $\eta$ assumed to be zero in Proposition 1 but Gaussian in Proposition 2? The decision is confusing, and the specific form seems to be chosen solely to make the proof go through.
  - CRLB results are standard and also apply equally to additive input-dependent perturbations. The same Fisher-information calculation would hold for $x' = x + p(x) + \eta$ , meaning the bounds are not specific to multiplicative attacks. Thus, the CRLB-based proposition does not establish a unique advantage for multiplicative attacks and only restates that parameter estimation is harder in the presence of noise.

- Finally, the theoretical analysis appears weakly connected to the practical method. The proposed implementation does not rely on an $\ell_2$ constraint but instead uses perceptual loss. The idea of erasing fingerprints by using methods like sampling, transformations, and Gaussian blurring is already explored in prior works such as TraceEvader and StatAttack [1], and given the above concerns, the presented theory does not seem to explain the empirical success of the proposed attack. The rationale behind the design remains unclear. Although the paper claims to require no access to any AMs, the design in fact embeds many human priors about AMs into the attack process.

- The authors wrote "Moreover, using a network to invert images imprints its fingerprints onto the recovered content, further degrading defense efficacy". If this is true, then a trivial attack would be to simply pass the generated image through any unrelated image-to-image model, which would already remove the fingerprint.

- The citation format is incorrect. When citing a work rather than the authors, the paper should use the ``\citep{}`` command (which produces "(Ouyang et al., 2024)") rather than ``\cite{}`` command (which produces "Ouyang et al. (2024)").

Ref:

[1]: Hou et al., Evading DeepFake Detectors via Adversarial Statistical Consistency. CVPR 2023.

**Questions:**

- Why is $w_i \in \\{0.5, 0.3, 0.2\\}$? Why are $\beta_1, \beta_2, \beta_3$ set as $\\{0.5, 0.1, 0.4\\}$? What happens when these values are changed?
- I also do not understand why the threat model restricts the attacker to having "no access to or information about any AMs". In realistic scenarios, attackers can easily access various AMs as their papers and codes are mostly opensourced. If access to some AMs could enable the attacker to design stronger attacks, then that would be definitely be better.

---

> ### Author Response · Authors · 2025-12-01
> **Response to Reviewer Spzx（part1）**
>
> We sincerely thank the reviewer for their thorough and insightful comments, which have helped us significantly strengthen the paper. Below, we provide a point-by-point response to each of the reviewer’s concerns..
>
> Response to Weakness 1:
>
> We agree with the reviewer that the algebraic equivalence is trivial. However, our core contribution is not about algebraic form—it lies in the attack intent, mechanism, and effect.
>
> Our work introduces a new attack paradigm: instead of merely obscuring fingerprints like existing additive methods, we eliminate them at their source by exploiting a key structural property: model fingerprints are not independent noise, but content-coupled modulations arising from content-dependent operations such as up-sampling, which manifest as structured modulation.
> Additive attacks perturb it without disrupting its underlying generation mechanism, leaving forensic traces intact and easily defensible (Sec. 3.3). In contrast, our multiplicative attack directly targets the modulation process itself. By leveraging the structural prior and optimizing the inductive bias for elimination, we disrupt the fingerprint’s structural coupling with the image, thereby achieving genuine elimination. While our theoretical analysis (Theorem 1, Proposition 2) uses an L2 norm constraint on Δx for analytical tractability, its purpose is to establish the existence of such a multiplicative operator—not to frame the attack as additive.
>
> Empirical evidence further confirms this distinction:
> Frequency domain: As shown in Fig. 1, additive attacks (e.g., TraceEvader) preserve high-frequency spectral patterns, whereas our method fundamentally reshapes them—direct visual evidence of fingerprint elimination.
> Residual correlation: Our residual Δx exhibits high Pearson correlation with the original image (PCC ∈ [0.5, 1]; Fig. 5, right), characteristic of multiplicative operations. Additive attacks yield near-zero correlation (PCC∈[0,0.25]).
> Residual magnitude: The L2 distance of our residuals varies significantly across images (Fig. 5, left), reflecting content-dependent perturbation. Additive attacks produce stable, low-magnitude residuals.
>
> We acknowledge that the original manuscript could better articulate this structural insight. In the revised version, we have significantly strengthened Section 4.1, where we explicitly introduce the “Structural Prior from Content-Coupled Fingerprints” and explain how it motivates our design. Our framework embeds this prior as an inductive bias, guiding the adversarial model to learn fingerprint elimination—not confusion.
>
> In summary, while the algebraic form x' = x + Δx is mathematically equivalent, the structure and origin of Δx are fundamentally different. Our multiplicative attack is not simply "additive in disguise"; it is a content-aware, structurally targeted attack designed to exploit the generative mechanism of fingerprints. This alignment with the fingerprint’s intrinsic structure enables true fingerprint elimination, yielding superior performance against both attribution models and defensive mechanisms, as demonstrated empirically.
>
> Response to Weakness 2:
>
>  We agree that Theorem 1 establishes existence rather than quantifying attack effectiveness—and this is by design.
> The purpose of Theorem 1 is to provide a formal foundation for the multiplicative attack paradigm. It proves that, under mild conditions (differentiability and non-degeneracy of the attribution model), there exists a content-dependent adversarial matrix W that can evade F while satisfying the distortion budget. This justifies our core design choice: fingerprint elimination via multiplication, not addition.
>
> The reviewer rightly points out that Theorem 1 does not bound the magnitude of the change in the attribution decision or the degree of fingerprint removal. We acknowledge this limitation and explicitly state in the revised manuscript that Theorem 1 is only a starting point. This theoretical grounding allows us to move beyond heuristic methods and build a principled framework for fingerprint elimination.  The quantitative performance, including attack success rate, visual fidelity, and the actual degree of fingerprint elimination, is not derived from the theorem itself but is the direct result of our subsequent framework design and extensive experimentation.
>
> In summary, Theorem 1 serves as a minimal but necessary theoretical guarantee that enables our subsequent method design and empirical validation.

---

> ### Author Response · Authors · 2025-12-01
> **Response to Reviewer Spzx（part2）**
>
> Response to Weakness 3:
>
> We agree that the original presentation of the noise term η was confusing, and we have clarified this in the revised manuscript. The noise η is not introduced by the attacker. Rather, it models real-world measurement uncertainty that any defender inevitably faces—such as interpolation artifacts during pre-processing. Its purpose is to enable a standard statistical analysis (via the Cramér–Rao Lower Bound) of the defender’s estimation performance under realistic conditions.
>
> Crucially, even in the idealized noise-free case (η=0), the inversion problem remains ill-posed and non-identifiable: as shown in Proposition 1, given only a single observed pair (x, x'), there exist infinitely many (\tilde{x}, \tilde{W}) such that \tilde{x} \odot \tilde{W} = x′. Thus, the non-invertibility is inherent to the multiplicative structure itself, not an artifact of noise.
>
> Moreover, while the CRLB is a general tool, its implications are uniquely severe for our attack due to the content-coupled nature of the residual Δx=x⊙(W−1). Unlike additive perturbations (which are typically designed to be independent of x, Figure 10), our residual is intrinsically modulated by the image content. This coupling renders the Fisher information matrix highly ill-conditioned, leading to extremely large estimation variance, even with paired data.
>
> Critically, our experiments in Section 5.3 empirically validate this theoretical limitation: even when defenders have access to paired clean/adversarial images and train powerful inversion models (DnCNN, Autoencoder), they fail to recover the original fingerprint, and the attack success rate remains above 97.68%. This confirms that the statistical non-invertibility of the multiplicative attack translates directly into practical defense resistance.
>
> Response to Weakness 4:
>
> Our method is indeed motivated by the structural prior of fingerprints rather than the human prior.  We firstly propose the multiplicative attack, which leverages the structural prior as an explicit inductive bias, directly disrupting this modulation mechanism through a multiplicative operation with an adversarial matrix.
>
> However, directly optimizing a fixed matrix W per image is neither scalable nor generalizable.  To address this, we parameterize W as an input-dependent function and implement it using a neural network. This preserves the multiplicative structure while enabling efficient, data-driven learning.
>
> Our framework is designed to explicitly instill the inductive bias into the adversarial model, enabling it to effectively eliminate traceable fingerprints (optimized by L_{spatial} and L_{spectral}) while preserving visual fidelity (optimized by L_{perceptual}). Critically, we use sampling and transformation not as direct perturbations (as in TraceEvader), but as tools to synthesize artificial fingerprints from real data, enabling the model to learn genuine fingerprint elimination without requiring any DeepFakes or access to AMs.
>
> Our quantitative analysis in Section 5.4 validates (Figure 5) this design, the strong correlation between residuals and original images  and the variable L2 magnitude distribution of perturbations confirm the multiplicative nature of our attack, confirming that the learned adversarial network faithfully realizes the theoretical design.
>
> To better highlight the principled link between theory and practice, we have revised the objective in Section 4.3.2 and added a concluding sentence that explicitly links the framework design to the theoretical insight on input-dependent adversarial matrix.
>
> Response to Weakness 5:
>
> We clarify that simply passing an image through an arbitrary image-to-image model does not eliminate the original generative model’s fingerprint. While such a process may imprint new artifacts (i.e., the reconstructor’s own “fingerprint”), it does not remove the content-coupled modulation left by the source generative model—precisely what attribution models rely on.
>
> In fact, our paper already evaluates this exact scenario under the Regeneration attack (Table 1), which reconstructs DeepFakes using a diffusion model (SD v2.1) to “sanitize” them. Despite introducing new network-induced artifacts, Regeneration achieves only 78.60% average ASR, significantly lower than our method’s 97.08%. This gap confirms that mere reconstruction or reprocessing cannot eliminate the original fingerprint; only a method that explicitly eliminates the traceable fingerprint can achieve genuine untraceability.
> Thus, the effectiveness of our attack stems not from “any network,” but from deliberate inductive bias design that enables the model to erase, rather than overlay, traceable signals.

---

> ### Author Response · Authors · 2025-12-01
> **Response to Reviewer Spzx（part3）**
>
> Response to Weakness 6:
>
> We have corrected all citations to use \citep{} throughout the manuscript.
>
> Response to Weakness 7:
>
> These values were not chosen arbitrarily, but are empirically optimized to achieve two objectives: fingerprint elimination and visual fidelity. To rigorously justify this choice, we conducted a comprehensive ablation study (Section 5.4, Figure 4), where we fix \beta_1 = 0.5,  and vary \beta_2, \beta_3 across multiple configurations.
>
> The results show that the setting (\beta_1, \beta_2, \beta_3) = (0.5, 0.1, 0.4) achieves the highest attack success rate while maintaining excellent perceptual quality, outperforming all alternative configurations. Thus, this configuration represents the optimal trade-off: it enables genuine fingerprint elimination while keeping distortions visually imperceptible.
>
> Response to Weakness 8:
>
> The black-box and AM-agnostic threat model is not an artificial limitation, but a deliberate design choice that enables universal applicability and practical robustness. Training an attack against known AMs (e.g., via white-box optimization) risks overfitting, leading to poor transferability. By eliminating the underlying fingerprint itself rather than optimizing against a specific AM, our method achieves universal evasion across all AMs without any access or adaptation. As shown in Table 1, our attack achieves 97.08% average ASR across 6 heterogeneous AMs, demonstrating strong generalization beyond the scope of white-box or transfer-based approaches.

---

### Official Review · Reviewer_kscU · 2025-10-30

**Soundness:** 1
**Presentation:** 4
**Contribution:** 1
**Rating:** 2
**Confidence:** 5

**Summary:**

This paper attempts to develop a defense strategy against attribution attacks. First, it reviews the limitations of existing approaches, which mainly focus on adding perturbations to images without addressing the removal of model fingerprints, the key indicators used for model attribution. Then, the authors introduce a framework for generating multiplicative perturbations, which constitutes the main claimed contribution of this work. However, the paper’s novelty appears limited, and its motivation and overall logic are not clearly articulated. The anonymous code provided in abstract is dead. Further details are discussed below.

**Strengths:**

The overall writing quality is clear and well-organized, making the paper easy to follow. The visual demonstrations are also well-designed and effectively support the presented ideas, helping readers better understand the proposed method and its outcomes.

**Weaknesses:**

The primary claim of this paper is that existing approaches depend solely on additive perturbations applied to images, which are insufficient to remove the underlying model fingerprints responsible for attribution. To overcome this limitation, the authors propose an adversarial network incorporating multiple constraint loss terms that consider spatial, spectral, and perceptual aspects.

However, my main concern is that these constraints appear to focus on controlling visual distortions rather than directly addressing the removal of model fingerprints. While the paper provides some theoretical justification, it is not strong enough to convincingly substantiate the central claim.

In addition, the experimental section does not include attribution accuracy results on clean images, making it difficult to assess the inherent challenge of the task or the true effectiveness of the proposed approach. This omission substantially undermines the paper’s contribution. Moreover, although the authors highlight the strength of their method in black-box defense settings, the corresponding results are missing from the table, which further adds to the confusion and raises doubts about the completeness of the evaluation.

Overall, the motivation and contribution of this work remain unconvincing, and in its current form, the paper does not meet the standards expected for acceptance at ICLR.

**Questions:**

NA

---

> ### Author Response · Authors · 2025-12-01
> **Response to Reviewer kscU （part1）**
>
> Thank you very much for your thorough review and valuable feedback. We sincerely appreciate the time and effort you have dedicated to evaluating our work. We would like to address your concern as follows:
>
> Response to Reviewer Core Contribution and Novelty：
>
> The reviewer’s comment that our paper “aims to develop a defense strategy against attribution attacks” reflects a fundamental misunderstanding. Our work is not a defense method; rather, it proposes a novel and more powerful attack methodology. The core contribution lies in the attack, not in defense.
>
> Specifically, firstly, our analysis and preliminary experiments reveal a critical limitation of existing attribution attacks: they are easily defended against and fail to eliminate the fingerprint, as they merely obscure the fingerprint.
>
> In contrast, we propose a multiplicative attack that fundamentally eliminates the model-specific fingerprint, thereby achieving truly untraceable DeepFakes. The novelty of our work manifests in both theory and methodology:
> (1) Theoretical novelty: We theoretically identify that the multiplicative attack can provably eliminate GMs' fingerprints by leveraging their content-coupled modulations, and prove this attack is statistically non-invertible, rendering it intrinsically evasive against AMs, even under defenses.
> (2) Methodological novelty: We propose a universal and black-box multiplicative attack method which instills the inductive bias into the adversarial model, enabling it to effectively eliminate traceable fingerprints while preserving visual fidelity without requiring any DeepFakes or access to AMs.
>
> We acknowledge that the original manuscript could more clearly articulate this structural insight. In the revised version, we have significantly strengthened the motivation in Section 4.1, where we explicitly introduce the “Structural Prior from Content-Coupled Fingerprints”. This insight directly motivates our design: the multiplicative operation disrupts the fingerprint’s modulation mechanism, and our framework embeds this principle as an inductive bias to guide effective fingerprint elimination.
>
>
> Response to Reviewer’s Concern on Loss Design and Fingerprint Elimination：
>
> Our framework is fundamentally designed as an end-to-end pipeline around the core objective of model fingerprint elimination, not merely visual distortion control. Specifically, the data synthesis module is designed to generate fingerprint-mimicking images from real data, and the model construction module trains the adversarial network on these synthetic pairs to eliminate artifacts while preserving visual fidelity with the guidance of a multi-domain loss function.
>
> The multi-domain loss function is explicitly engineered to target fingerprint elimination across complementary domains. The perceptual loss maintains semantic integrity while permitting necessary modifications to eliminate fingerprints. While the spatial and spectral losses directly confront the fingerprint, they directly suppress the low-level artifacts within the pixel domain and high-frequency patterns within the frequency domain, respectively.
>
> Our ablation study (Section 5.4， Table 1) provides compelling evidence of this design's necessity. For example, removing Lspectral causes ASR to plummet to 80.31\%, and against the Reverse attribution model, it drops dramatically to 50.04\%, confirming that these constraints are essential for fingerprint elimination rather than mere visual preservation.
>
> In direct response to the reviewers’ comments, we have revised the manuscript to clarify (i) the role of the multi-domain loss design, and (ii) the interplay among the individual modules in our framework. Specifically, we updated the framework diagram (Figure 2) and completely rewrote Section 4.3.2 (Overview) to explicitly describe the data-flow and dependencies between modules. Additionally, in the Model Construction subsection, we added detailed explanations of each loss term, demonstrating how they jointly suppress fingerprints in the spatial and frequency domains while preserving semantic content.

---

> ### Author Response · Authors · 2025-12-01
> **Response to Reviewer kscU （part2）**
>
> Response to Reviewer's concern on the original performance of AMs and the black-box defense results:
>
> We sincerely appreciate the reviewer’s valuable feedback, pointing out the omission of baseline results and the ambiguity in our black-box defense presentation. We have added Table 4 in the Appendix to report the attribution accuracy on clean images and have referenced this data in the main text.
>
> Furthermore, we apologize for the confusion caused by the original Table 2 regarding the black-box defense results. We clarify that this experiment is designed to evaluate the effectiveness of our method against AMs that are enhanced using adversarial samples from other attack methods. To resolve the misleading presentation, we have removed Table 2 and replaced it with Figure 3, which provides a clearer visualization of our method's performance against these defended models. We have also revised the figure caption to explicitly define the x-axis and the specific experimental settings to ensure clarity.

---

### Official Review · Reviewer_8PRM · 2025-11-01

**Soundness:** 3
**Presentation:** 3
**Contribution:** 3
**Rating:** 6
**Confidence:** 3

**Summary:**

This paper proposes a universal, black-box multiplicative attack to make DeepFakes untraceable by eliminating model fingerprints rather than merely confusing them with additive perturbations. The method trains an image-to-image network on real images augmented with sampling and transformation operations to mimic generative fingerprints; at test time it applies a pixelwise multiplicative mapping to DeepFakes to suppress attribution cues while preserving perceptual quality. Experiments across 12 GMs and 6 attribution models report high attack success and robustness against adversarial training and inversion defenses.

**Strengths:**

1. Clear articulation of why additive attacks preserve fingerprints; frequency-domain evidence and defensive degradation back this claim.
2. Simple but clever training using sampling/transform ops to mimic GM artifacts without GM access, which is broad applicability.

**Weaknesses:**

Theory makes local (first-order) arguments and per-pixel noise assumptions. Some modern AMs often include non-linear, patchwise, or frequency pipelines and heavy pre-processing, have you tested its performance on such models?

**Questions:**

See weaknesses.

---

> ### Author Response · Authors · 2025-11-27
> **Response to Reviewer 8PRM**
>
> We sincerely thank you for your thoughtful and constructive feedback. We greatly appreciate your positive assessment of our work’s soundness, presentation, and contribution, as well as your insightful concern regarding the theoretical assumptions in our analysis.
>
> You rightly point out that Theorem 1 relies on a first-order (local linear) approximation of the attribution model $F(x)$, and that the defense difficulty analysis assumes i.i.d. per-pixel Gaussian noise. These are standard simplifying assumptions commonly adopted in theoretical analyses of adversarial robustness. Their purpose here is to establish the theoretical feasibility of multiplicative attacks: such an attack can exist and is inherently difficult to invert under mild regularity conditions.
>
> Crucially, our method makes no assumptions about the internal structure, architecture, or even the existence of the target attribution model during training or inference. It is a completely AM-agnostic, black-box framework. By eliminating the generative model's fingerprint within DeepFakes, our method ensures that the resulting adversarial image no longer contains any information about its source generative model. Consequently, all attribution models, regardless of their internal pipeline (patch-wise, frequency-based, or non-linear) fail to trace the origin, not because they are “fooled,” but because there is nothing left to trace.
>
> This design is validated experimentally on a diverse set of modern AMs, many of which embody the complexities you mentioned, including non-linear mappings, patch-wise operations, frequency-domain processing, and heavy pre-processing:
> (1) DNA-Det (patch-wise contrastive learning),
> (2) DCT (pure frequency-domain analysis),
> (3) Reverse and LTracer (non-linear inverse modeling),
> (4) POSE (open-set, highly non-linear decision boundary).
>
> As shown in Table 1, our method achieves an average ASR of 97.08\% across all six state-of-the-art attribution models, including 98.56\% against DNA-Det (patch-wise) and 100.0\% against DCT (frequency-domain), precisely the types of models you highlighted. This empirical success demonstrates that the practical effectiveness of our approach transcends the idealized assumptions used in the theoretical analysis.
>
> In essence, the theory provides a principled motivation that multiplicative attacks can provably eliminate fingerprints under mild differentiability conditions, while the experiments validate robustness in real-world settings where AMs are highly non-linear, patch-based, or frequency-aware.
>
> Thank you again for your valuable comment.

---

### Official Review · Reviewer_Sam5 · 2025-11-07

**Soundness:** 3
**Presentation:** 3
**Contribution:** 2
**Rating:** 4
**Confidence:** 3

**Summary:**

This paper presents a universal black-box attack method for untraceable deepfakes, that trains an adversarial model solely using real data and applicable for various generative models and agnostic to attributiom models. The experimental results show that the attack success rate of the proposed method is not only high across a wide range of attribution models, but also effective against defense mechanisms.

**Strengths:**

1.[effectiveness] The proposed method achieves better attack performance than the previous methods, with some loss on SSIM and LPIPS.

**Weaknesses:**

1.[clarity] In Figure 1, only GAN models are analyzed for the spectral property. Do diffusion models show any similar property? This paper also includes diffusion models apart from GANs, so the diffusion model cannot be missed in this figure.

2.[clarity] This method is effective. However, as a black-box method, its efficiency is also important for discussion. How does the proposed method compare with the previous methods in terms of compute?

3.[ablation] The proposed method balances three parameters in L352 (please also add equation numbers -- this is a typesetting issue). They control perceptual, spatial, and spectral loss functions. In Table 1, I noted that the proposed method's SSIM and LPIPS results are far from the best. If we tune the parameters to improve the SSIM and LPIPS to match with TraceEvador, how does the proposed method perform? Namely, if the proposed method still outperforms TraceEvador at the same SSIM/LPIPS trade-off, the proposed method is strongly better. Otherwise we do not have enough evidence to claim this as a real improvement. At the mean time, the impact of the three parameters is not discussed in this paper.

**Questions:**

See weaknesses.

---

> ### Author Response · Authors · 2025-11-27
> **Response to Reviewer Sam5**
>
> We sincerely thank the reviewer for your insightful comments, which have helped us improve the clarity and completeness of our work. We address your concerns as follows:
>
>  \textbf(Response to Q1} :
>
>   While Figure 1 in the main text originally focused on GAN-based models for clarity, we did include comprehensive spectral analyses of diffusion models (DMs) in the appendix:
>   (1) Figure 7 presents frequency-domain comparisons for DeepFakes generated by Stable Diffusion v1.5 and v2.1, both before and after applying our attack.
>  (2) Figure 15 further extends this analysis to three state-of-the-art diffusion models（FLUX, PixArt, and SD3）， demonstrating consistent and significant spectral alterations after the attack, analogous to the GAN case.
>
>  These results collectively confirm that model fingerprints，whether left by GANs or DMs are effectively eliminated by our method, leading to comparable spectral changes across both model families.
>
>   In direct response to the reviewer’s suggestion, we have revised Figure 1 in the main paper to include representative examples from both GANs (e.g., InfoMaxGAN, StarGAN, StyleGAN3) and diffusion models (e.g., SD v1.5, SD v2.1). All newly added content in the updated figure is highlighted in blue for ease of reference.
>   We believe this enhancement strengthens the paper’s clarity and better reflects the universality of our approach across both GANs and diffusion models.
>
>
> Response to Q2:
>
> Our method is a highly efficient universal black-box attack: once the adversarial model is trained , it can be applied to any DeepFake image via a single forward pass, without requiring access to or interaction with any attribution model. This means one inference = evasion of all AMs, which is highly efficient in real-world black-box scenarios.
>
> To better highlight the practical efficiency of our method, we have added a computational efficiency evaluation in Section 5.4. The evaluation reports end-to-end inference time for generating 20,000 adversarial images under identical experimental conditions. Our approach completes this task in 60.6 seconds, achieving a 12× speedup over TraceEvader (732.7 seconds), which underscores its practical efficiency.
>
>
> Response to Q3:
>
> The current parameter setting achieves the best balance between fingerprint elimination and image quality, yielding the highest attack success rate without compromising perceptual fidelity.
> Our method fundamentally operates by eliminating model fingerprints, which inherently requires more substantial modifications than adding perturbations. Despite this, visual quality remains comparable to state-of-the-art methods.
>
> To better contextualize this trade-off, we have added an ablation study in Section 5.4 (Figure 4), evaluating the impact of $\beta_2,\beta_3$ with the $\beta_1$ fixed at 0.5. The configuration ($\beta_1,\beta_2,\beta_3$)=(0.5,0.1,0.4) achieves the highest ASR (97.1\%) without compromising image fidelity. While our approach introduces slightly more distortion than TraceEvader, it still surpasses many existing methods in perceptual fidelity, striking a favorable balance between evasion strength and visual integrity.
>
>
> Thank you again for your valuable comment.

---

### Meta-Review · Area_Chair_MAb7 · 2026-01-06

**Summary:**

Key concerns raised by reviewers:
- Clear distinction against additive attacks (Spzx), or just passing through any image-to-image model (Spzx)
- Lack of evaluation on clean images as a baseline of task difficulty (kscU)
- Other technical issues: observed degradation of visual quality vs eg. TraceEvader (), and diffusion models not included in the spectral property analysis in Fig-1 -- actually included in the appendix (Sam5)

**Reviewer Concerns:**

Authors adequately addressed seemingly all reviewer concerns, many of which were confusions possibly due to presentation issues or the inherent novelty of the approach.  Some questions were already addressed in appendices or introduced during the rebuttal -- through significant amendments to the main text.

Overall, the paper tackles a challenging problem in DeepFake detection with deep insights into what makes attacks successful and how to work with generative models (in an agnostic fashion) to evade defenses (also in an agnostic fashion), supported by theoretical derivations and detailed analyses (eg. spectral properties).  Experimental results validate the claims, showing improvements over SOTA, and the codes are provided.  This makes for a valuable contribution to ICLR this year.

**Reviewer Scores:**

Initial ratings came as 6/4/2/2.  While it's unclear how the reviewers may go about updating their ratings, and taking into account significant initial confusions with thorough treatment by the authors, I'd expect a final rating closer to 6.

---

### Decision · Program_Chairs · 2026-01-26

Accept (Poster)